# SPARC: Separating Perception And Reasoning Circuits for Test-time Scaling of VLMs

**Niccolo Avogaro** [*1 2]  **Nayanika Debnath** [*1 2]  **Li Mi** [1]  **Thomas Frick** [2]  **Junling Wang** [1]  **Zexue He** [3]  **Hang Hua** [3]  **Konrad Schindler** [1]  **Mattia Rigotti** [2]

## Abstract

Despite recent successes, *test-time scaling*—dynamically expanding the token budget during inference as needed—remains brittle for vision-language models (VLMs). Unstructured visual reasoning chains entangle perception and reasoning, leading to long, disorganized contexts where small perceptual mistakes may cascade into completely wrong answers. Reasoning also requires expensive reinforcement learning with hand-crafted rewards. Here, we introduce SPARC (Separating Perception And Reasoning Circuits), a modular framework that explicitly decouples visual perception from reasoning. Inspired by sequential sensory-to-cognitive processing in the brain, SPARC implements a two-stage pipeline where the model first performs explicit visual search to localize question-relevant regions, then conditions its reasoning on those regions to produce the final answer. This separation enables independent test-time scaling with asymmetric compute allocation (e.g., prioritizing perceptual processing under distribution shift), and supports selective optimization (e.g., improving the perceptual stage alone when it is the bottleneck for end-to-end performance). It also accommodates compressed contexts by running global search at lower image resolutions and allocating high-resolution processing only to selected regions, thereby reducing visual token count and compute. SPARC outperforms monolithic baselines and strong visual-grounding approaches across challenging visual reasoning tasks, such as improving Qwen3VL 4B on the $V^*$ VQA benchmark by 6.7 points and surpassing "thinking with images" by 4.6 points in an OOD setting with a $200\times$ lower token budget.

---

[*]Equal contribution  [1]ETH Zürich  [2]IBM Research  [3]MIT-IBM Watson AI Lab. Correspondence to: Niccolo Avogaro <niccolo.avogaro1@ibm.com>.

*Proceedings of the $43^{rd}$ International Conference on Machine Learning*, Seoul, South Korea. PMLR 306, 2026. Copyright 2026 by the author(s).

## 1. Introduction

Multimodal Vision-Language Models (VLMs) have become the de facto standard in visual reasoning and perception (Li et al., 2025). VLMs are architectures that combine visual and textual inputs. By aligning a vision backbone with an LLM (Huang et al., 2023), they extend the impressive NLP capabilities of LLMs to the vision realm (Alayrac et al., 2022; Chen et al., 2023b; Hua et al., 2025; Li et al., 2023; Chen et al., 2023a; Liu et al., 2023; Zhu et al., 2024; Peng et al., 2024; Achiam et al., 2023; Karlinsky et al., 2025). Among the capabilities that VLMs inherit from LLMs is Chain-of-Thought (CoT) reasoning (Wei et al., 2022), a test-time compute mechanism to iteratively generate the output step-by-step, which can be optimized via Reinforcement Learning and has been popularized by models like ChatGPT-o1 (OpenAI, 2024) and DeepSeek-R1 (Guo et al., 2025). Works like ViGoRL (Sarch et al., 2025) and Deep-Eyes (Zheng et al., 2026) have demonstrated that multimodal chain-of-thought reasoning, obtained by interleaving pure text CoT reasoning with image content, can be explicitly grounded to the relevant visual evidence in the image via a multi-turn workflow that calls appropriate image analysis tools. In this so-called "thinking with images" paradigm first introduced in the OpenAI ChatGPT-o3 report (OpenAI Research, 2025), the model alternates between reasoning steps and perceptual actions (like selecting a region of interest in the image). Such grounded multi-modal CoTs can yield significantly better performance in visual reasoning tasks, especially when it comes to high-resolution perception where one must repeatedly focus attention on small but decisive image details.

A core issue of "thinking with images", and multi-modal CoT reasoning in general, is that learning is considerably more complex than for standard, text-only reasoning: the LLM must acquire the ability to manage multi-turn conversations and tool calls that repeatedly mix visual and reasoning tokens within the context window (Sarch et al., 2025; Su et al., 2025a; Zheng et al., 2026; Kumar et al., 2025). This is not only computationally expensive, but also more brittle, particularly for smaller models whose performance rapidly degrades when faced with long token sequences due

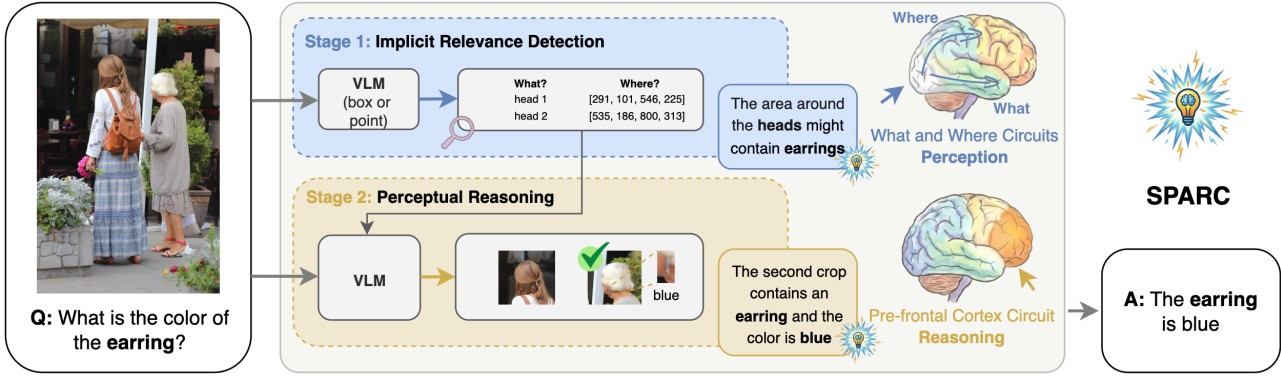

*Figure 1.* Overview of the SPARC framework. We decouple the VLM inference process into two distinct functional circuits. Stage 1 (Perception): The What and Where Circuits perform Implicit Relevance Detection (IRD), taking the image and question as input to output relevant crop coordinates (e.g., localizing the woman's ear). Stage 2 (Reasoning): The "Prefrontal Cortex Circuit" synthesizes a CoT by reasoning over the high-resolution crops identified in the first stage and outputs the final answer ("blue"). This separation enables independent optimization and robust, efficient test-time scaling.

to visually heavy contexts and extended reasoning chains (Tian et al., 2026), which amplify VLMs' difficulty with fine visual details (Rahmanzadehgervi et al., 2024) and their tendency to fall into bias-driven "mirage reasoning" (Vo et al., 2026; Asadi et al., 2026). Furthermore, a monolithic approach is inflexible and lacks a mechanism to adapt the allocated compute to the difficulty of the vision task: when to terminate the response is left to the LLM.

Here, we propose a new, more efficient test-time scaling strategy for VLMs, motivated by context engineering principles (Mei et al., 2025) that maintain that operating CoTs in an unstructured fashion (in our case, entangling perception and reasoning tokens), hinders effective organization of the context and can thereby impair task performance.

Our architecture draws inspiration from systems and visual neuroscience, and specifically the biological brain's hierarchical information processing architecture, where early visual areas first extract low-level features that are elaborated through parallel "what" and "where" visual pathways (Mishkin et al., 1983; Kravitz et al., 2011). This information then converges and is mixed in high-dimensional codes in prefrontal cortex (Rigotti et al., 2013; Tye et al., 2024), the cortical area viewed as responsible for integrating sensory inputs and contextual information, and supporting the implementation of our flexible goal-oriented thoughts, learning and behavior (Miller & Cohen, 2001; Sung et al., 2025).

Based on this view, we propose a two-stage pipeline as illustrated in Figure 1. Given a visual question and its associated answer, we do not directly prompt the model to return the answer, but rather ask it to find the relevant image content. Image crops detected by this Implicit Relevance Detection (IRD) are then used to re-prompt the model for an answer to the actual question, given the relevant image regions. That prompting strategy, by itself, turns out to lead to better

results than native "thinking with images"; moreover, we show that it has a number of interesting properties.

First, when using the two-step pipeline with an efficient initial IRD step it becomes possible to scale perception at test time independently from reasoning. As an example, employing self-consistency over eight roll-outs of the IRD step, with a shared $KV$-cache, creates only a few additional text tokens and an additional crop, but boosts performance of the full pipeline by up to 9.3%. Second, the two steps can be trained separately, without forgetting the model's generic, pretrained capabilities. For instance, when training for the particular perception needs of some technical domain, there is no danger of losing the ability to reason and produce CoTs. Third, dedicated training for the IRD task is extremely efficient in terms of both data and training time, because one needs to rollout only a small number of tokens for the crop coordinates instead of going through long multi-modal multi-turn reasoning chains in every iteration.

In summary, our contributions are:

- We introduce SPARC, an effective prompting scheme that enables reliable *test-time scaling of perception tasks, in zero-shot mode and with very small computational overhead*.

- We show that SPARC enables *asymmetric compute allocation between perception and reasoning*, allowing a targeted self-consistency mechanism that *scales more favorably than naive ensembling*.

- We demonstrate that decoupling visual reasoning into separate perception and reasoning stages enables *efficient training of the perception model without degrading the reasoning model's original capabilities*.

## 2. Related Work

**Vision-Language Models and Grounding**. Recent advancements in Vision-Language Models (VLMs) have primarily focused on extending Large Language Models (LLMs) with visual perception capabilities, typically via a visual encoder (e.g., CLIP (Radford et al., 2021), SigLIP (Zhai et al., 2023)) connected to the LLM backbone through a projection layer (Liu et al., 2023; Li et al., 2024; Bai et al., 2023). Modern architectures like LLaVA-OneVision-1.5 (An et al., 2025), MM1 (McKinzie et al., 2024), and Qwen3-VL (Bai et al., 2025) have scaled this paradigm, improving resolution handling and multi-image reasoning. While most VLMs output text only, a critical evolution is the integration of fine-grained visual grounding, enabling models to output spatial coordinates (bounding boxes or points) alongside text. Models such as Kosmos-2 (Peng et al., 2024), Qwen3-VL, and PaliGemma (Beyer et al., 2024) natively support this grounding capability, treating coordinates as text or special tokens. More recent works like GLaMM (Rasheed et al., 2024) and Molmo2 (Clark et al., 2026) further refine this by interleaving segmentation or point-based grounding with reasoning, establishing a strong paradigm where precise spatial localization is intrinsic to the generation process (Cho et al., 2025; Wang et al., 2025c), achieving a surprising level of performance (Avogaro et al., 2025), sometimes even emerging from attention maps (Zhang et al., 2025a), despite having limits when it comes to extremely fine-grained localization (Zhang et al., 2026).

**Test-Time Scaling of Large Language Models**. The paradigm of test-time scaling—allowing autoregressive models to generate additional intermediate tokens before outputting a final answer—has emerged as a powerful, training-free method for enhancing performance. Foundational techniques such as Chain-of-Thought (CoT) (Wei et al., 2022) and Self-Consistency (Wang et al., 2023) demonstrated that linear reasoning traces significantly improve problem-solving capabilities. Subsequent works expanded this into non-linear structures, such as Tree of Thoughts (ToT) (Yao et al., 2023) and Graph of Thoughts (GoT) (Besta et al., 2024), which enable more deliberate exploration of the solution space. More recently, this extensive reasoning capability has been baked directly into models via strong post-training techniques. Systems like OpenAI o1 (OpenAI, 2024) and DeepSeek-R1 (Guo et al., 2025) utilize Reinforcement Learning (RL) with sparse rewards to encourage the model to autonomously verify and refine its internal chain of thought. Furthermore, recent studies indicate that such sophisticated reasoning patterns can also be induced in a training-free manner, for instance through training-free Group Relative Policy Optimization (GRPO) (Cai et al., 2025) or by eliciting reasoning via external cognitive tools (Ebouky et al., 2025).

**Test-time Scaling of Vision-Language Models**. A growing body of work adapts "R1-style" test-time scaling to VLMs through "thinking with images" (Su et al., 2025b), which interleaves and entangles intermediate visual operations (e.g., zooming, cropping, pointing) with textual CoTs. This paradigm has been mostly developed through reinforcement learning frameworks that incentivize explicit visual reasoning. For instance, Point-RFT (Ni et al., 2025) and ViGoRL (Sarch et al., 2025) utilize reinforcement fine-tuning to align reasoning traces with precise grounded spatial references. Similarly, DeepEyes (Zheng et al., 2026) and Pixel Reasoner (Su et al., 2025a) employ curiosity-driven or specific reward structures to encourage models to actively query visual data during the reasoning process, with some work such as (Kumar et al., 2025) mostly focusing on efficiency. Beyond static images, Video-R4 (Tang et al., 2025) extends this concept to video understanding.

## 3. Test-Time Scaling of Perception

Detailed visual perception is a prerequisite for a wide array of applications, ranging from document understanding and outdoor robotics to satellite image analysis. Test-time scaling has emerged as the primary paradigm to boost the performance of both LLMs and VLMs. Allowing the model to generate additional tokens prior to the final answer during inference enables it to reference a broader range of contextual information during reasoning. This mechanism has been shown to consistently enhance both predictive accuracy and robustness (Wei et al., 2022; OpenAI, 2024; Guo et al., 2025). In the present work, we focus on the perceptual abilities of VLMs. Our goal is to enhance the model's handling of task-relevant visual input at test time in an efficient and robust manner.

Intuitively, more detailed image understanding necessitates a richer visual representation. We posit that this translates directly to an increased number of image tokens at test time. This intuition aligns with the recently popular "thinking with images" approach to VLM test-time scaling. In that framework, models generate extended reasoning traces that are interleaved with *zoom-in* actions, i.e., the model invokes a tool to retrieve relevant image crops in higher resolution.

While we agree with the high-level concept of zooming into image locations that may be important for the task, we argue that for tasks that are primarily perceptual, lengthy intermediate text generation and complex multi-turn handling are superfluous and can even be counter-productive. The open-form traces produced by such a procedure bring little benefit for the actual perception task; on the contrary, they contradict context engineering principles that suggest to employ structured and modular composition (Mei et al., 2025) and can promote excessively long traces resulting in hallucinations (Diao et al., 2026). Instead, we propose a

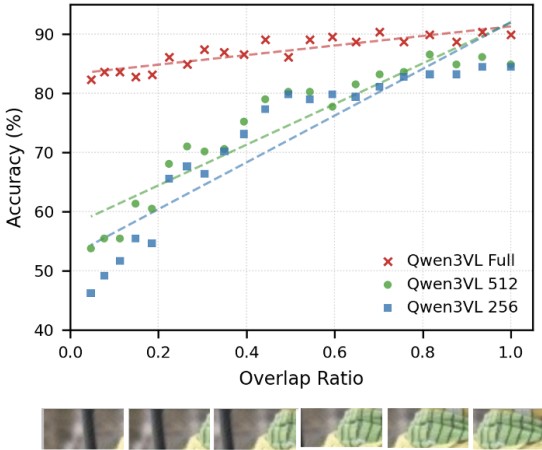

**Question:** What is the color of the **scarf**?
**Answer:** The color of the scarf is green.

*Figure 2.* The plot shows downstream reasoning accuracy against the crop overlap ratio. While performance generally degrades as overlap decreases, this effect is most pronounced for lower resolutions. Crucially, at high overlap ratios, the 256px model converges to the performance of the full-resolution model. This demonstrates that accurate perceptual guidance can fully compensate for the loss of global visual detail, allowing for highly efficient inference.

shift towards **visual context engineering**: our hypothesis is that a well-structured context that contains *only* the necessary high-resolution image content offers a more compact and robust perceptual representation than long, unorganized multi-modal roll-outs.

### 3.1. Experimental Setup

In order to validate the intuition that a well-structured prompt is enough to solve hard perception tasks, we run experiments on the $V^*$ benchmark (Wu & Xie, 2024), a standard testbed for *thinking with images*. Featuring high-resolution images that contain small objects, $V^*$ requires a model to find and inspect objects and to resolve complex spatial relationships. We select that benchmark because its main difficulty is detailed visual perception: the hardest task is to locate small objects and look at them at a sufficient resolution. Once one has zoomed in on the relevant visual region, providing the correct answer is straightforward.

For our investigation, we prompt a VLM to solve the VQA task on the $V^*$ benchmark. As the benchmark comes with ground truth image locations, we modify the input prompts to include image crops with varying degrees of misalignment to the ground truth and measure the impact of that misalignment on VQA performance. Results for the off-the-shelf version of Qwen3-VL 4B (Bai et al., 2025) are shown in Figure 2. See Appendix A.1 for experimental results conducted with Molmo2 (Clark et al., 2026).

We systematically modulate the usefulness of the crops through controlled spatial perturbations. For each target object, we generate a crop with the same height and width as the ground truth bounding box, but shift its center by a distance $r$ relative to the true location. By progressively reducing $r$ from the half-diagonal of the bounding box down to zero, we generate a set of images that partially overlap the ground truth crop, and use those figures as visual prompts. Figure 2 illustrates the model's VQA accuracy as a function of the overlap. We repeat the experiment with varying token budget by resizing images to smaller resolutions (longer side 256, respectively 512 pixels, keeping the original aspect ratio). The purpose of the latter comparison is to verify whether a more accurate localization can compensate for a lower resolution.

### 3.2. Findings

We observe that supplying the model with increasingly precise crops drives performance up towards the theoretical upper bound. In other words, if one had an oracle to guide the selection of the bounding box within the image, then that would be sufficient to unlock the model's existing reasoning ability. Furthermore, our results point to an important efficiency trade-off: if a model operating at 256 pixel resolution achieves even a modest localization accuracy (20% overlap with the ground truth), it already surpasses a 512px model without object localization, at a fraction of the computational cost. This effect is most pronounced in extreme low-resolution regimes. In other words, sufficient performance is achieved if small crops are positioned accurately, whereas high resolution over a larger context does not seem to bring much benefit.

Together, the empirical findings suggest an inference scheme in which perceptual computations are offloaded to specialized modules: on the one hand, perception is *unsurprisingly* important for visual reasoning, but on the other hand, it can evidently run as a fairly independent low-level process, thus minimizing the burden on the reasoning backbone. We note this layout mirrors the layout of biological brains, where a dedicated perceptual stream (the occipital lobe) processes visual stimuli, whereas high-level reasoning about the visual input is left to a separate region (the prefrontal cortex).

## 4. Two-Stage Architecture: Decoupling Perception and Reasoning

Building on the neuroscience motivation that perception and reasoning are distinct cognitive faculties, we propose decoupling these processes in VLMs. To operationalize this, we implement a sequential two-step prompting protocol. In the first phase (Relevance Detection), the model acts as a perceptual circuit, strictly tasked with localizing salient image regions. In the second phase (Perceptual Reason-

ing), the model serves as a reasoning circuit, generating the final answer conditioned on the extracted regions. Employing sequential prompting stages exercises the *context engineering* best practice of managing the context window via structured, modular composition (Mei et al., 2025), and steers the model towards sampling from two distinct output distributions (Xie et al., 2022; Min et al., 2022), effectively activating separate *functional circuits* for visual search and logical deduction, rather than entangling them.

Separating distinct relevance detection and reasoning operations intuitively enables the model to focus on addressing each specific demand. The *relevance detection step*, for instance, demands that the model localize pertinent image regions based on a specific query. Unlike standard Referring Expression Comprehension (REC) (Mao et al., 2016; Yu et al., 2016), where the target is explicitly named (e.g., 'find the red ball'), this objective requires the model to infer latent visual relevance from a high-level reasoning prompt. Consequently, defining an 'optimal' crop becomes an ill-posed problem: strictly speaking, the ideal crop is not necessarily the tightest bounding box around an object, but rather the visual window that maximizes the probability of a correct prediction in the subsequent reasoning step. While some queries demand precise object detection, others benefit from looser crops that preserve contextual relationships—a phenomenon we analyze in Appendix A.2. Given these distinct requirements, we term this task Implicit Relevance Detection (IRD).

*Table 1.* In-domain (ID) and Out-of-distribution (OOD) average performance of SPARC. The ID metric is computed as the mean over $V^*$, HRBench-4K and HRBench-8K, while OOD is computed as the average over the XLRS remote sensing benchmark.

| Method | ID Average | | | OOD Average | | |
|---|---|---|---|---|---|---|
| | 256 | 512 | Full | 256 | 512 | Full |
| *Qwen3-VL 4B* | | | | | | |
| Native Performance | 41.7 | 48.8 | 72.6 | 46.2 | 48.4 | 53.5 |
| "Thinking w/ images" | 36.8 | 52.2 | 73.1 | 43.1 | 48.3 | 48.3 |
| SPARC *(Ours)* | **51.0** | **60.6** | **74.8** | **48.7** | **52.9** | **54.8** |
| *Qwen3-VL 8B* | | | | | | |
| Native Performance | 41.4 | 49.4 | 79.0 | 47.5 | 47.9 | 53.2 |
| "Thinking w/ images" | 40.9 | **56.5** | 78.1 | 44.0 | 48.0 | 50.1 |
| SPARC *(Ours)* | **45.4** | 54.8 | **79.5** | **52.4** | **53.3** | **56.0** |
| *Molmo2 4B* | | | | | | |
| Native Performance | 48.1 | 53.2 | 60.8 | 38.1 | 39.1 | 39.9 |
| "Thinking w/ images" | — | — | — | — | — | — |
| SPARC *(Ours)* | **48.7** | **57.0** | **62.9** | **40.8** | **42.1** | **43.2** |
| *Molmo2 8B* | | | | | | |
| Native Performance | 45.8 | 52.4 | 57.8 | 37.5 | 39.4 | **39.1** |
| "Thinking w/ images" | — | — | — | — | — | — |
| SPARC *(Ours)* | **47.4** | **55.0** | **59.1** | **39.1** | **39.9** | 39.0 |

### 4.1. Experimental Setup

We evaluate our method across two distinct model families, selected to represent two different spatial grounding modalities: Qwen3-VL, which performs relevance detection via *bounding box generation* (Bai et al., 2025), and Molmo2, which uses *point-based detection* (Clark et al., 2026).

For *Molmo2*, a squared $256 \times 256$ crop resolution is extracted from the point. We evaluate at different token budgets: full resolution, and downsized at 512 and 256 longest image size. In this case, the crops are taken from the original full-size image and downsized accordingly if exceeding the target resolution, in order to avoid hacking. The evaluation is conducted at 4B and 8B sizes. Appendix A.11 presents the results for the larger Qwen3-VL 235B-A22B model. We employ a two-step prompting technique. We first instruct the model to strictly output the coordinates (or points) of image regions relevant to the query. In the second step, we re-prompt the model with the image content and append the newly generated crops from the first step. Prompts are available in Appendix A.3.

We present our comprehensive quantitative analysis in Table 1. This evaluation aggregates performance across high-fidelity perception benchmarks, specifically $V^*$ and the high-resolution suites HRBench-4K and HRBench-8K (Wang et al., 2025b) with in-domain (ID) settings. Additionally, we report robustness results on the XLRS remote sensing suite (Wang et al., 2025a), with extremely large and high-resolution remote sensing images. We treat XLRS as an out-of-distribution (OOD) proxy, given that remote sensing imagery is scarce in standard instruction-tuning corpora, typically dominated by documents, UIs, and natural images. Consequently, this benchmark offers a challenging evaluation of the model's ability to generalize its cropping mechanism to non-standard visual domains. We employ greedy decoding to ensure deterministic evaluation and minimize variance, particularly given the multiple-choice nature of the datasets. We benchmark our decoupled method against the state-of-the-art "thinking with images" paradigm, which is natively supported by the Qwen3-VL architecture. In this case, we use the off-the-shelf generation parameters. The full tables of results are provided in Appendix A.13 and a comparison with prior art can be found in Appendix A.4.

To rigorously quantify the trade-off between efficiency and accuracy, we conduct a granular analysis of SPARC's performance across varying computational budgets. Specifically, we evaluate Qwen3-VL 4B on the $V^*$ benchmark under a spectrum of input resolutions. By mapping these configurations against their respective inference costs, we construct the Pareto frontier reported in Figure 3. We further analyze real-world inference costs in Appendix A.8.

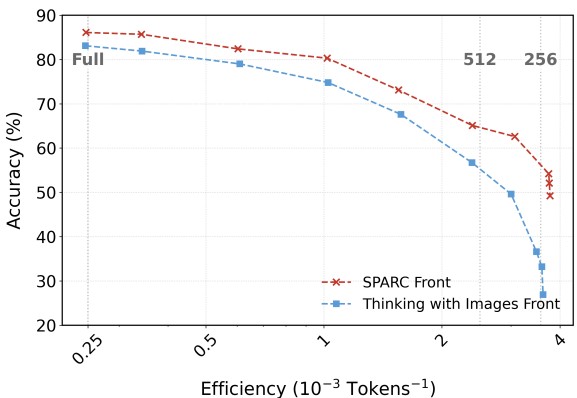

*Figure 3.* SPARC outperforms the "thinking with images" paradigm of Qwen3-VL 4B, providing a more robust and efficient inference paradigm. This advantage is particularly pronounced in perceptually demanding scenarios, where SPARC achieves superior localization and reasoning with significantly fewer tokens.

### 4.2. Findings

Our results demonstrate that we can significantly enhance model performance in a completely training-free manner. Notably, this approach not only enables effective perception scaling for the Molmo family but also surpasses the native "thinking with images" paradigm in Qwen3-VL. We observed consistent performance gains across all benchmarks, with SPARC consistently outperforming its native baseline. Furthermore, we present targeted studies in Appendices A.7 and A.12 showing that SPARC has no detrimental influence in less perception-oriented benchmarks, and does not suffer from "tunnel vision", i.e. context collapse when dealing with queries requiring global or relational understanding. Regarding specific architectures, we observe that Molmo2-8B underperforms relative to its 4B counterpart, a phenomenon consistent with findings reported in the original work. We note that we experienced unstable behavior with the Qwen3-VL family of models: in our experiments, sometimes "thinking with images" does not even reach the native model performance. We provide the full quantitative analysis in Appendix A.13 and a qualitative analysis of failure modes in Appendix A.14.

The advantages of our methodology are most pronounced in low-resolution regimes. In these settings, the generated crops do not merely provide token redundancy; they actively restore critical high-frequency visual information lost during downsizing, effectively bypassing the perceptual bottlenecks of the base resolution. This effect is evident in the Pareto frontier, but it is even more striking in the OOD remote sensing evaluation. On XLRS, for instance, Molmo2 operating at 256-pixel resolution with crops surpasses the performance of the standard model prompted at full resolution. This represents a paradigm shift in efficiency: given the dataset's average dimension of $8500 \times 8500$ pixels, our method achieves superior accuracy while processing approximately 0.1% of the visual tokens required for a naive full-resolution forward pass.

SPARC minimizes latency by sharing visual KV-caches between the two steps and truncating context to avoid the quadratic costs of entangled chains of thought. This decoupling unlocks asymmetric test-time scaling, enabling dynamic compute allocation, such as enforcing consistency between predicted crops. Furthermore, it facilitates independent optimization, allowing the perceptual circuit to be fine-tuned without retraining the reasoning backbone. We dedicate the following sections to rigorously benchmarking these capabilities, showing how modularity allows for a more scalable and data-efficient VLM paradigm.

## 5. Scaling via Perceptual Consistency

A distinct advantage of our disentangled architecture is the ability to allocate inference budgets asymmetrically. While ensemble methods like Self-Consistency (Wang et al., 2023) are known to enhance performance by aggregating multiple rollouts, they typically incur a linear increase in total compute. In contrast, SPARC permits applying self-consistency selectively to the perception branch. Crucially, this yields a unique efficiency property: because the perception module outputs simple coordinates in token space, generating multiple detection hypotheses is computationally inexpensive. By aggregating these lightweight rollouts via standard bounding-box fusion algorithm, we can construct a single, high-confidence visual context for the reasoning step. Consequently, the expensive reasoning backbone processes only one refined input, avoiding the prohibitive cost of running $N$ full-chain reasoning traces.

As illustrated in Figure 2, we observe a sharp performance cliff: accuracy degrades rapidly as the intersection with the ground truth decreases. This confirms that maximizing spatial coverage of the target region is a strict prerequisite for correct reasoning. Motivated by this, we propose a strategy that prioritizes recall over precision during the crop aggregation phase. By merging redundant overlapping proposals while retaining distinct non-overlapping regions, we ensure that the model maximizes its effective receptive field over the relevant features, even at the cost of including marginally more background context. We offload the detection of these noisy crops to the reasoning step.

### 5.1. Experimental Setup

We perform $N$ independent inference IRD rollouts (e.g., $N = 8$) on the global image using a non-zero temperature ($T = 0.7$). This encourages the model to explore diverse localization hypotheses, which are then aggregated using Weighted Boxes Fusion (WBF). Unlike standard Non-

Maximum Suppression (NMS) which simply discards proposals, WBF computes a weighted average of overlapping predictions to derive a consensus bounding box. We merge bounding boxes with at least 50% intersection over union. Distinct, non-overlapping bounding boxes are retained and forwarded directly to the reasoning stage. Results for $N = 4$ and $N = 8$ can be found in Table 2.

*Table 2.* Performance gains and average crop counts using Weighted Box Fusion (WBF) with $N = 4$ and $N = 8$ rollouts. The method allows for effective test-time scaling by refining crop proposals in the text space, drastically reducing the volume of image tokens processed during the final reasoning phase of SPARC.

| | Average | | | Crops Number | | |
|---|---|---|---|---|---|---|
| **Method** | **256** | **512** | **Full** | **256** | **512** | **Full** |
| *Qwen3-VL 4B* | | | | | | |
| SPARC | 51.0 | 60.6 | 74.8 | 1.59 | 1.63 | 1.64 |
| SPARC WBF 4 | 54.2 | 65.4 | 81.7 | 2.38 | 2.05 | 1.72 |
| SPARC WBF 8 | **55.7** | **67.0** | **82.0** | 3.30 | 2.54 | 1.88 |
| *Qwen3-VL 8B* | | | | | | |
| SPARC | 45.4 | 54.8 | 79.5 | 1.23 | 1.31 | 1.46 |
| SPARC WBF 4 | 48.9 | 63.4 | 80.9 | 1.96 | 1.77 | 1.66 |
| SPARC WBF 8 | **49.9** | **64.1** | **81.1** | 2.54 | 2.19 | 1.82 |
| *Molmo2 4B* | | | | | | |
| SPARC | 48.7 | 57.0 | 62.9 | 1.96 | 1.62 | 1.72 |
| SPARC WBF 4 | 48.7 | 57.8 | 63.3 | 3.53 | 2.76 | 2.73 |
| SPARC WBF 8 | **52.6** | **58.3** | **64.1** | 5.57 | 4.09 | 4.03 |
| *Molmo2 8B* | | | | | | |
| SPARC | 47.4 | 55.0 | **59.1** | 1.55 | 1.63 | 1.54 |
| SPARC WBF 4 | 47.9 | 54.9 | 58.2 | 2.57 | 2.22 | 2.14 |
| SPARC WBF 8 | **48.0** | **55.9** | 58.3 | 3.76 | 3.15 | 2.92 |

## 5.2. Findings

The results in Table 2 demonstrate that enforcing consistency in bounding box generation is a robust strategy for enhancing test-time performance. Across all evaluated models, we observe a monotonic increase in accuracy as the number of initial rollouts ($N$) grows from 1 to 8. This confirms that the stochastic aggregation of multiple perceptual hypotheses effectively denoises the localization step, leading to more reliable visual contexts for the downstream reasoning task.

A key advantage of our Weighted Box Fusion (WBF) approach is its ability to improve accuracy without a proportional linear increase in downstream computational cost. While we initiate 8 independent rollouts during the relevance detection phase, the de-duplication mechanism ensures that the final number of crops forwarded to the reasoning module remains significantly lower. For instance, with Qwen3-VL 4B at 256 resolution, employing 8 rollouts results in an average of only 3.30 final crops. This highlights the efficiency of our asymmetric scaling: we gain the benefits of broad exploration in the cheap perceptual space while maintain-

ing a lean context for the expensive reasoning phase. A qualitative analysis on how the WBF merging combines predictions at different resolutions can be found in Appendix A.14.

An interesting trend emerges when analyzing the relationship between input resolution and crop count. As the input image size increases (from 256 to Full resolution), the average number of final crops consistently decreases. We hypothesize that at higher resolutions, the model's ability to solve the Implicit Relevance Detection (IRD) task improves, leading to higher confidence and greater consensus among the $N$ rollouts. Consequently, the WBF algorithm merges these highly overlapping predictions into fewer, more unified bounding boxes. This suggests that as perceptual fidelity improves, the model naturally converges on the correct region, reducing the need for extensive ensemble de-duplication. Inspired by bottom-up visual search approaches in the literature, we provide a bottom-up SPARC test-time scaling analysis in Appendix A.5 and further diagnostics over the WBF behavior in Appendix A.6.

## 6. Fine-Tuning for Pure Perception

Complementary to test-time scaling, performance can be enhanced by shifting the computational burden to the training phase. By explicitly training the VLM to execute IRD more robustly, we can directly improve accuracy on downstream VQA tasks.

Our objective is to enhance the model's perceptual capabilities. However, naively fine-tuning a VLM on Implicit Relevance Detection (IRD) risks *catastrophic forgetting*, effectively degrading its reasoning performance, a phenomenon we demonstrate empirically in Appendix A.10. SPARC's disentangled architecture offers a solution: because perception and reasoning occur in distinct steps, we can optimize them independently. We implement this by training a specialized Low-Rank Adaptation (LoRA) module exclusively for the detection phase. At test time, this adapter is dynamically activated only during perceptual search, ensuring improved localization accuracy without compromising the integrity of the reasoning backbone.

A distinct advantage of this modular approach is its training simplicity. Unlike the "thinking with images" paradigm—which necessitates complex reinforcement learning frameworks, custom reward shaping, and extensively curated process-supervision datasets—our method relies on standard supervised fine-tuning of a lightweight LoRA. This allows us to inject specialized perceptual capabilities without the engineering overhead or instability associated with inducing latent reasoning traces in monolithic models.

## 6.1. Experimental Setup

Training our explicit perception modules requires a VQA dataset enriched with spatial relevance annotations. As we pointed out in Section 4, IRD correctness is not based on training on a unique label, but instead a prediction is considered correct if it leads to a correct answer. In order to obtain such annotations, we perform a round of synthetic data generation on the DeepEyes dataset (Zheng et al., 2026), tailoring the annotation format to the grounding modality of each target architecture:

- **Bounding-Box Annotations (Qwen3-VL):** We utilize the large-scale Qwen3-VL 235B-A22B model, leveraging its native "thinking with images" capabilities. We extract the crop coordinates generated during the model's intermediate tool calls and apply rejection sampling—retaining only those traces that yield a correct final answer. This process results in a high-quality dataset of approximately 23,000 samples.

- **Point-Based Annotations (Molmo2):** For the Molmo family, we employ the 8B variant (the largest publicly available model at the time of writing). We execute the two-step inference pipeline described in Section 4 to generate relevance points. Following the same filtering protocol as for Qwen3-VL, we retain only the successful traces, yielding a curated dataset of approximately 14,000 samples.

We perform Supervised Fine-Tuning (SFT) for two epochs using a standard autoregressive next-token prediction objective. Crucially, we conduct this training across three distinct resolution scales to evaluate the necessity of high-fidelity inputs for the detection task. This experimental design is motivated by our findings in Section 4, where the base models demonstrated high baseline proficiency in the IRD task at full resolution. We hypothesize that training exclusively at native resolution may render the optimization task too trivial, potentially inducing overfitting due to a lack of sufficient difficulty. Moreover, training at full resolution would mean performing pure knowledge distillation of the bigger model, which is a much weaker training signal than trying to solve the same task at lower resolution. More details about the training setup are in Appendix A.9.

## 6.2. Findings

As detailed in Table 3, fine-tuning the explicit perception module yields systematic performance improvements across all evaluated dimensions, spanning diverse model families, parameter counts, and prompting strategies. The sole exception is Molmo2-4B, where the fine-tuned model performs comparably to the baseline at lower resolutions. We attribute this plateau to a distillation bottleneck: the synthetic

*Table 3.* We compare the SPARC baseline against specialized adapters fine-tuned at varying resolutions. Counter-intuitively, the model trained on the lowest resolution (SPARC SFT 256, highlighted) achieves the highest accuracy across most test settings. This supports the hypothesis that low-resolution training acts as a regularizer: by forcing the model to infer relevance from coarser signals, it learns more robust perceptual features than models trained via trivial high-resolution distillation.

| | Average Score | | |
|---|---|---|---|
| **Method** | **256** | **512** | **Full** |
| *Qwen3-VL 4B* | | | |
| SPARC | 51.0 | 60.6 | 74.8 |
| SPARC SFT Full | 50.8 | 58.6 | 75.6 |
| SPARC SFT 512 | 51.0 | 59.1 | 74.6 |
| SPARC SFT 256 | **51.7** | **64.0** | **76.8** |
| *Qwen3-VL 8B* | | | |
| SPARC | 45.4 | 54.8 | 79.5 |
| SPARC SFT Full | 46.3 | 55.5 | 80.4 |
| SPARC SFT 512 | 44.4 | 57.1 | 80.9 |
| SPARC SFT 256 | **53.1** | **64.3** | **82.4** |
| *Molmo2 4B* | | | |
| SPARC | **48.7** | **57.0** | 62.9 |
| SPARC SFT Full | 46.9 | 52.7 | 61.4 |
| SPARC SFT 512 | 47.2 | 53.4 | 61.0 |
| SPARC SFT 256 | 48.2 | 56.8 | **63.8** |
| *Molmo2 8B* | | | |
| SPARC | 47.4 | 55.0 | 59.1 |
| SPARC SFT Full | 47.8 | 54.2 | 57.5 |
| SPARC SFT 512 | **48.5** | 53.2 | 59.3 |
| SPARC SFT 256 | 48.1 | **55.8** | **59.6** |

dataset was generated using a relatively weak Molmo2-8B teacher, which likely failed to provide sufficiently high-quality supervision for the 4B student. We hypothesize that employing a stronger teacher for data generation would resolve this limitation and unlock further gains. Nevertheless, with this single exception, our results confirm that base models—despite their strong zero-shot capabilities—benefit significantly from targeted optimization for the Implicit Relevance Detection task.

Our resolution ablation study reveals a counter-intuitive but favorable result: training at reduced image resolutions is not only computationally cheaper but also more effective than full-resolution training. This validates our hypothesis that training on the high-resolution task is prone to overfitting. When trained at native resolution, the model faces a trivial optimization path, easily mimicking the teacher model without developing a deep understanding of visual relevance. By artificially degrading the input resolution, we increase the task difficulty, forcing the model to rely on structural and semantic context rather than perfect memorization. This constraint prevents the optimization from collapsing into shallow distillation, ensuring that the learned perceptual circuit is robust and capable of generalization.

# 7. Conclusion

In this work, we introduced SPARC, a systems neuroscience inspired inference framework that explicitly decouples visual perception from reasoning in VLMs. By separating region localization from question answering, SPARC prevents perceptual errors from cascading into hallucinations, significantly reduces computational overhead, and enables asymmetric test-time scaling by aggregating lightweight visual searches. Furthermore, this modularity allows for data-efficient targeted fine-tuning of the perceptual circuit without degrading the base model's pre-trained reasoning capabilities. A current limitation of SPARC is its reliance on VLMs with native spatial grounding ability (e.g., coordinate or point outputs): applying this framework to purely text-generative VLMs would require integrating a dedicated external object detection model. Looking ahead, SPARC's decoupled architecture provides a robust foundation for more complex visual tasks and opens several promising avenues for future research. Potential directions include exploring iterative zooming pipelines, or extending the framework to video inputs, where a lightweight perceptual circuit could perform spatio-temporal object tracking to isolate salient clips. Additionally, downstream reasoning accuracy could be further maximized by applying targeted visual augmentations or super-resolution techniques directly to the extracted crops prior to the reasoning phase.

## Acknowledgements

We thank the anonymous reviewers for their constructive and thoughtful comments. The authors also extend their gratitude to Dr. Rogerio Feris for making this collaboration possible, and to Prof. April Wang for her support and guidance. Financial support is gratefully acknowledged from Horizon Europe grant No. 101213369 (DVPS) supported by the Swiss State Secretariat for Education, Research and Innovation (SERI) under contract number 25.00138 to Konrad Schindler and Li Mi; Horizon Europe grant No. 101070408 (SustainML) supported by the Swiss State Secretariat for Education, Research and Innovation (SERI) under contract number 22.00295 to Niccolo Avogaro and Mattia Rigotti; and from ETH AI Center through an ETH AI Center doctoral fellowship to Junling Wang.

## Impact Statement

This paper presents work whose goal is to advance the field of Machine Learning. There are a number of potential societal consequences of our work, none of which we feel needs to be specifically highlighted here.

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

# A. Appendix

## A.1. Test-Time Scaling Generalization to Molmo2 Architecture

To verify that the *resolution compensation* phenomenon is not specific to the Qwen3 architecture, we replicate our crop overlap ablation using the Molmo2-4B (Clark et al., 2026) family. As shown in Figure 4, we observe an identical behavioral pattern: while the performance of downscaled models (256px, 512px) drops precipitously when crop alignment is poor, it recovers dramatically as the Intersection-over-Union (IoU) with the ground truth increases. When provided with oracle-level crops (Overlap Ratio ≈ 1.0), the 256px and 512px baselines effectively close the performance gap with the full-resolution model, offering a computationally efficient alternative to processing high-resolution images. This validates our core premise: investing compute in precise localization (via SPARC) allows us to offload the heavy reasoning step to much lighter, low-resolution inference passes without sacrificing accuracy.

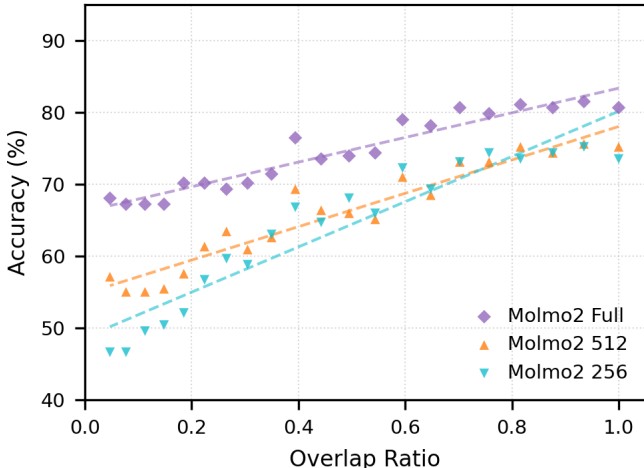

*Figure 4.* We extend our analysis to the Molmo2 architecture, plotting accuracy against crop overlap ratio. Consistent with our findings on Qwen3-VL, the low-resolution variants exhibit a steep performance recovery as crop precision improves. Notably, high-quality crops allow the efficient 256px and 512px models to approach the performance upper bound of the Full-resolution baseline, further supporting the motivation of the SPARC pipeline.

## A.2. Implicit Relevance Detection as an Ill-Posed Problem

To empirically determine the optimal field of view, we evaluate performance while progressively upscaling the ground truth crop size by a factor of up to $10\times$ (Figure 5). Initially, we observe a consistent performance gain across all resolutions as the crop expands (e.g., peaking around scale $2.5\times$ for the 256px model), validating that strictly tight bounding boxes often exclude necessary semantic context. However, a critical trade-off emerges for the resolution-constrained variants (256px and 512px). Since these crops are resized to fit a fixed pixel buffer (e.g., max 256px), excessively enlarging the physical crop region forces aggressive downsampling, diluting the object's visual fidelity. Consequently, while the Full-resolution model remains robust at large scales, the 256px model suffers a sharp performance collapse beyond scale $4\times$, as the loss of high-frequency detail outweighs the benefit of added context.

## A.3. Prompts

We provide the specific prompts employed for the Implicit Relevance Detection (IRD) phase (Step 1) and the subsequent Reasoning phase (Step 2) for both the Qwen and Molmo architectures. Empirically, we observed that the single most critical factor for ensuring robust instruction adherence was the explicit definition of the output format. Enforcing a strict structural constraint—specifically, requesting JSON output for Qwen and Point coordinates for Molmo—significantly reduced syntax errors and hallucinations compared to less constrained prompts.

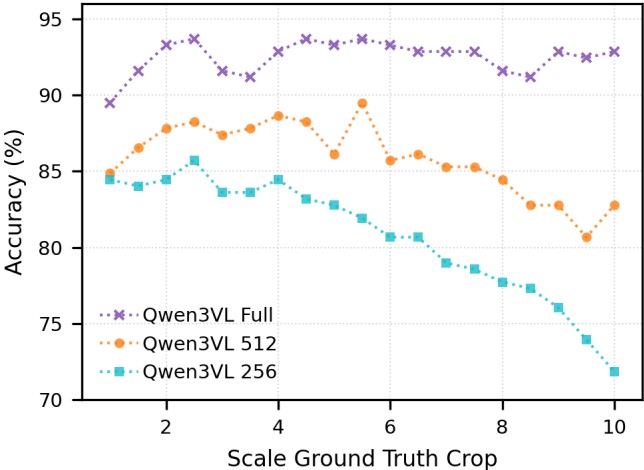

*Figure 5.* We measure reasoning accuracy as a function of crop expansion factor (up to $100\times$ the original box area). While moderate expansion (scales $2\times$–$4\times$) improves performance by providing necessary context, excessive scaling leads to a sharp decline for resolution-constrained models.

---

**IRD Qwen3-VL Prompt**

```
{image}
You are a helpful assistant capable of doing object detection.  You will
be given an image and a question for context.  Your role is not to answer
the question, but identify the objects the user will ask for and return
their 2D bounding box and label in JSON format.  The images will be very
low resolution, but the objects will be there.  Given this image and the
following question:
{question}
DO NOT ANSWER THE QUESTION. Identify the relevant objects and return their 2D
bounding box and label in JSON format.
```

---

**QA Qwen3-VL Prompt**

```
{image}
{crop}×N
You are a helpful assistant.  You are given an image and relevant crops to
answer the following question:
Question:  {question}
Answer with the option's letter from the given choices directly.  Predict the
letter only.
```

---

**IRD Molmo2 Prompt**

```
{image}
Question:  {question}
You are a helpful assistant.  Your task is to POINT to the objects relevant to
the user's question.
```

---

**QA Molmo2 Prompt**

```
{image}
{crop}×N
You are a helpful assistant.  You are given an image and relevant crops to
answer the following question:
Question:  {question}
Answer with the option's letter from the given choices directly.  Predict the
letter only.
```

---

### A.4. Comparison with "Thinking with Images" Approaches on V*

Table 4 compares SPARC (using Qwen3-VL 8B) against state-of-the-art scaling and RL-based approaches. Notably, our fully optimized pipeline (*w/ SFT*) achieves 91.2%, narrowly surpassing the biggest model of the family Qwen3-VL 235B-A22B (91.1%) and significantly outperforming sophisticated RL baselines like DeepEyes (90.1%) and ViGoRL-7B (86.4%). This result confirms that explicitly disentangling perception allows an 8B model to rival architectures with $30\times$ more parameters, suggesting that the primary performance bottleneck is often perceptual. Moreover, SPARC achieves these gains via stable Supervised Fine-Tuning (SFT) of the perception circuit, avoiding the instability and complexity of the reinforcement learning recipes required by competing methods.

*Table 4.* Performance of our proposed SPARC Framework compared to the existing "thinking with images" approaches in the literature. Metrics marked with * were reproduced by the authors.

| Model Name | V* |
|---|---|
| **"Thinking w/ images"** | |
| ChatGPT-o1 (OpenAI, 2024) | 69.7 |
| Reinforcing VLMs (Kumar et al., 2025) | 80.1 |
| Pixel-Reasoner (Su et al., 2025a) | 84.3 |
| ViGoRL-7B (Sarch et al., 2025) | 86.4 |
| DeepEyes (Zheng et al., 2026) | 90.1 |
| Qwen3-VL 235B-A22B (Bai et al., 2025) | 91.1* |
| ChatGPT-o3 (OpenAI Research, 2025) | 95.7 |
| **Qwen3-VL 8B** | |
| Native Performance | 87.0* |
| SPARC (Ours) | 88.7* |
|    w/ Consistency | 89.5* |
|    w/ SFT | 91.2* |

### A.5. Comparison with Bottom-Up Divide-and-Conquer Approaches

While SPARC introduces a highly efficient top-down architectural intervention to locate relevant regions before reasoning, it shares conceptual similarities with classical divide-and-conquer visual grounding methods. Standard divide-and-conquer approaches often rely on brute-force, bottom-up grid slicing to process high-resolution images. To empirically explore the structural trade-offs between these paradigms, we implemented a *bottom-up SPARC* variant. Specifically, we partition the input image into a $3 \times 3$ grid, process each of the 9 tiles through an independent IRD rollout to predict localized bounding boxes, and finally aggregate the local predictions back to global coordinates using WBF.

As demonstrated in Table 5, the bottom-up approach significantly improves performance at heavily constrained resolutions by isolating fine visual details within localized grids (e.g., Molmo2 4B improves from 48.7% to 62.3% at 256 resolution). However, this gain relies on a drastically higher computational overhead, processing up to $\sim$11 crops per image compared to SPARC's highly efficient $\sim$1.9 crops. Conversely, at higher resolutions, the original top-down SPARC framework remains vastly more efficient while achieving comparable or even superior peak accuracy (e.g., reaching 82.0% with Qwen3-VL 4B at Full resolution versus 81.3% for the bottom-up variant). These results confirm that while bottom-up slicing can

mitigate low-resolution perceptual bottlenecks, SPARC provides a strictly superior efficiency-accuracy trade-off when base resolutions are adequate, and can be easily integrated into existing test-time scaling pipelines.

*Table 5.* Performance and crop-efficiency comparison between standard top-down SPARC and a bottom-up $3 \times 3$ grid variant across different base models and resolutions. Metrics are averaged over $V^*$, HRBench-4K, and HRBench-8K.

| Method | Average | | | Crops Number | | |
|---|---|---|---|---|---|---|
| | **256** | **512** | **Full** | **256** | **512** | **Full** |
| *Qwen3-VL 4B* | | | | | | |
| SPARC | 51.0 | 60.6 | 74.8 | 1.59 | 1.63 | 1.64 |
| SPARC WBF 8 | 55.7 | 67.0 | **82.0** | 3.30 | 2.54 | 1.88 |
| Bottom-up SPARC | **61.0** | **68.0** | 81.3 | 5.51 | 5.60 | 5.92 |
| *Qwen3-VL 8B* | | | | | | |
| SPARC | 45.4 | 54.8 | 79.5 | 1.23 | 1.31 | 1.46 |
| SPARC WBF 8 | 49.9 | 64.1 | 81.1 | 2.54 | 2.19 | 1.82 |
| Bottom-up SPARC | **57.7** | **65.3** | **81.7** | 7.33 | 8.21 | 9.28 |
| *Molmo2 4B* | | | | | | |
| SPARC | 48.7 | 57.0 | 62.9 | 1.96 | 1.62 | 1.72 |
| SPARC WBF 8 | 52.6 | 58.3 | 64.1 | 5.57 | 4.09 | 4.03 |
| Bottom-up SPARC | **62.3** | **64.7** | **69.3** | 11.60 | 11.05 | 9.84 |
| *Molmo2 8B* | | | | | | |
| SPARC | 47.4 | 55.0 | 59.1 | 1.55 | 1.63 | 1.54 |
| SPARC WBF 8 | 48.0 | 55.9 | 58.3 | 3.76 | 3.15 | 2.92 |
| Bottom-up SPARC | **62.3** | **64.7** | **69.3** | 11.60 | 11.05 | 9.84 |

## A.6. WBF Scaling and Ablation

To analyze failure modes and the behavior of Weighted Box Fusion (WBF) under scaling, we conducted a diagnostic ablation on the $V^*$ dataset. We compared raw IRD rollouts against WBF deduplication, scaling up to 64 rollouts. The results are summarized in Table 6.

*Table 6.* WBF diagnostic ablation on the $V^*$ dataset comparing raw IRD rollouts versus WBF deduplication across varying rollout counts.

| Metric | Number of Rollouts | | | | |
|---|---|---|---|---|---|
| | **4** | **8** | **16** | **32** | **64** |
| *Raw Rollout* | | | | | |
| Union Area | 0.89% | 0.94% | 1.17% | 1.30% | 1.78% |
| Intersection | 0.19% | 0.19% | 0.19% | 0.19% | 0.20% |
| Accuracy | 88.2 | 87.8 | 88.2 | 89.0 | 88.0 |
| Crop Number | 5.4 | 10.5 | 21.3 | 42.6 | 85.4 |
| *WBF Deduplication* | | | | | |
| Union Area | 0.79% | 0.80% | 1.00% | 1.09% | 1.47% |
| Intersection | 0.18% | 0.18% | 0.18% | 0.18% | 0.18% |
| Accuracy | 88.2 | 89.5 | 88.7 | 89.1 | 88.6 |
| Crop Number | 1.57 | 1.68 | 1.93 | 2.09 | 2.44 |
| *Overall Reference* | | | | | |
| Ground Truth Area | 0.20% | 0.20% | 0.20% | 0.20% | 0.20% |
| Coverage Loss | 3.88% | 5.46% | 6.65% | 7.00% | 6.72% |

This ablation highlights several key takeaways regarding our region proposal efficiency:

- **High Localization Precision:** Target objects in $V^*$ are extremely small (Ground Truth Area is 0.20% of the image). Even with 64 naive rollouts yielding 85 total crops, the combined union area remains tightly bounded at 1.78%. This demonstrates that the IRD stage is highly precise and rarely diverts attention to irrelevant, far-off regions.

- **Accuracy Saturation:** WBF performance peaks at around 8 rollouts. At this stage, the intersection with the ground truth is already near-perfect (0.19% out of 0.20%). Adding crops beyond 8 does not harm performance through context distraction; rather, it simply offers no new visual information, resulting in stable but saturated accuracy.

- **Compression Efficiency:** WBF successfully compresses 85 raw crops (at 64 rollouts) down to just $\sim$2.4 highly relevant deduplicated crops. This mechanism retains maximum accuracy while maintaining strict context efficiency for the reasoning model.

### A.7. Robustness to Global Scene and Relational Queries

A common concern with crop-based architectures is the potential for "tunnel vision" or context collapse when facing queries that require global scene understanding or relational comparisons across distant objects (e.g., "Is the room empty?"). To stress-test these scenarios, we conducted a targeted analysis on two filtered dataset subsets:

- **$V^*$ Far Objects:** A subset of the $V^*$ benchmark strictly filtered for spatial relationship queries where the two target objects are separated by at least half the image's smallest dimension (16 QA pairs).

- **GQA Global:** A subset of the GQA (Hudson & Manning, 2019) test set (157 QA pairs) explicitly filtered using the "global" question tag, which encompasses whole-scene understanding.

*Table 7.* Performance on perception-heavy queries requiring global scene understanding or relational comparisons across distant objects (using Qwen3-VL 4B base model).

| Dataset | Method | Accuracy | |
|---|---|---|---|
| | | **256** | **Full** |
| **$V^*$ Far Objects** | Native Baseline | 62.5 | 68.8 |
| | SPARC | **68.8** | **75.0** |
| **GQA Global** | Native Baseline | 19.1 | 24.8 |
| | SPARC | **21.7** | **26.8** |

As demonstrated in Table 7, SPARC does not suffer from systematic failure modes or context collapse on these global queries. In fact, it strictly outperforms the native baseline across both evaluated resolutions. This robustness is fundamentally tied to SPARC's architectural design. During the reasoning stage, SPARC does not discard the original image; rather, the model is conditioned on *both* the localized high-resolution crop(s) and the original global image. If a query requires recognizing an empty room, the model naturally leverages the global image. If it requires comparing distant objects, the IRD stage either provides localized crops for both regions, or the reasoning module simply falls back to the global view. Because the global context is never lost, SPARC safely handles scene-level semantics without introducing structural penalties.

### A.8. Real-World Efficiency Metrics

We recognize the importance of measuring physical latency to capture real-world throughput. To provide a comprehensive view of total inference time, we conducted a server-side analysis comparing SPARC and "thinking with images" across Time to First Token (TTFT) and End-to-End (E2E) latency using the *vllm* (Kwon et al., 2023) inference service. We caution that both metrics remain brittle, implementation-dependent and subject to server load and framework optimizations. As detailed in Table 8, SPARC consistently achieves lower latency than the "thinking with images" baseline across all datasets and resolutions. Notably, because SPARC at 256 resolution outperforms "thinking with images" at full resolution in overall accuracy, our pipeline delivers superior performance while being over $200\times$ faster in TTFT and $50\times$ in E2E.

### A.9. Training Details

We fine-tuned the Qwen3-VL Instruct architecture using the Unsloth (Han et al., 2023) and TRL (von Werra et al., 2020) framework. To mitigate computational costs while maintaining performance, we employed LoRA finetuning across both the vision and language components of the model. Specifically, we applied LoRA adapters to the attention mechanisms, MLP modules, and vision encoders. Optimization was performed using the 8-bit AdamW optimizer, coupled with a linear learning rate scheduler. The training was executed using the TRL framework with gradient checkpointing enabled to support

*Table 8.* Comparison of Time to First Token (TTFT) and End-to-End (E2E) latency between SPARC and "thinking with images" (TwI).

| Dataset | TTFT (s) | | E2E Latency (s) | |
|---|---|---|---|---|
| | SPARC | TwI | SPARC | TwI |
| $V^*$ | | | | |
| 256 Resolution | 0.20 | 0.28 | 0.66 | 1.86 |
| Full Resolution | 2.97 | 3.05 | 4.77 | 6.12 |
| XLRS | | | | |
| 256 Resolution | 0.06 | 0.07 | 0.33 | 2.14 |
| Full Resolution | 13.19 | 13.51 | 13.57 | 15.86 |

longer context windows and higher batch sizes. Molmo2 was trained on the same framework using a standard HuggingFace Transformer implementation. Qwen3-VL 8B was trained on a single A100-80GB for approximately 12 hours, while Molmo2 naive implementation is more computationally expensive, requiring double the computation budget. Hyperparameters can be found in Table 9.

*Table 9.* Hyperparameter configuration for SPARC SFT fine-tuning.

| Category | Hyperparameter | Value |
|---|---|---|
| **Model Configuration** | Base Model | `unsloth/Qwen3-VL-8B-Instruct` |
| | Precision | 16-bit (LoRA) |
| | Max Context Length | 2048 |
| | Image Resolution | $256 \times 256$ |
| **LoRA Configuration** | Rank ($r$) | 16 |
| | Alpha ($\alpha$) | 32 |
| | Dropout | 0.0 |
| | Target Modules | Vision, Language, Attn, MLP |
| | Bias | None |
| **Optimization** | Optimizer | AdamW (8-bit) |
| | Learning Rate | $2 \times 10^{-4}$ |
| | Weight Decay | 0.001 |
| | Scheduler Type | Linear |
| | Warmup Steps | 100 |
| **Training Schedule** | Epochs | 5 |
| | Batch Size (per device) | 32 |
| | Gradient Accumulation | 4 |
| | Train/Val Split | 99% / 1% |

### A.10. Distilled SFT Baseline

To evaluate the efficacy of our decoupled training approach, we compare SPARC SFT against a standard Distillation SFT baseline, where the model is trained end-to-end directly on the teacher's reasoning traces using an identical tuning recipe. The evaluation reports average performance across the $V^*$, HRBench-4K, and HRBench-8K datasets.

As demonstrated in Table 10, the Distillation SFT approach generally underperforms the zero-shot "thinking with images" baseline, particularly at higher resolutions. This degradation highlights a critical vulnerability: standard SFT distillation on narrow, synthetic, multi-turn tool-calling data can induce *catastrophic forgetting*, effectively overwriting the strong reasoning priors established during the base model's extensive alignment phase.

Furthermore, due to the lengthy, interleaved multimodal chains of thought, training the Distillation SFT baseline consumes $4\times$ the GPU memory at a constant batch size compared to SPARC SFT. By strictly separating perception from reasoning, SPARC bypasses this degradation entirely. It allows for efficient optimization of spatial localization without corrupting the model's general reasoning capabilities.

*Table 10.* Performance comparison between zero-shot Thinking with Images, End-to-End Distillation SFT, and SPARC SFT. Results are reported as average accuracy across $V^*$, HRBench-4K, and HRBench-8K.

| Method | Average | | |
|---|---|---|---|
| | **256** | **512** | **Full** |
| *Qwen3-VL 4B* | | | |
| Thinking with Images | 36.8 | 52.2 | 73.1 |
| Distillation SFT | 37.7 | 50.2 | 66.6 |
| SPARC SFT | **51.7** | **64.0** | **76.8** |
| *Qwen3-VL 8B* | | | |
| Thinking with Images | 40.9 | 56.5 | 78.1 |
| Distillation SFT | 42.7 | 54.9 | 72.1 |
| SPARC SFT | **53.1** | **64.3** | **82.4** |

### A.11. Larger Model Scaling

To evaluate whether the architectural benefits of SPARC persist at the frontier scale, we extended our evaluation to the Qwen235B-A22B model on the $V^*$ dataset. As shown in Table 11, the results demonstrate that SPARC's decoupling of perception and reasoning scales effectively to large-scale models. These findings confirm that structured perceptual routing is not merely a compensatory mechanism for smaller models, but a persistent architectural enhancement that maximizes reasoning accuracy across all model scales.

*Table 11.* Performance evaluation of Qwen235B-A22B on the $V^*$ dataset across different input resolutions.

| Setting | $V^*$ | | |
|---|---|---|---|
| | **256** | **512** | **Full** |
| *Qwen3-VL 235B-A22B* | | | |
| Native Baseline | 47.5 | 52.1 | 89.5 |
| "Thinking w/ images" | 47.5 | 59.2 | 91.1 |
| SPARC | **53.0** | **65.1** | **91.6** |

### A.12. Performance on General Reasoning Tasks

To ensure that SPARC does not degrade capabilities on reasoning-dominated tasks, we evaluated it on the comprehensive MME Realworld (Zhang et al., 2025b) benchmark. As detailed in Table 12, SPARC maintains, and frequently improves, performance across broad reasoning categories when compared to the native baselines. For instance, at constrained resolutions (256 and 512), SPARC demonstrates noticeable gains in both perception and reasoning splits across various domains, including OCR and Diagram & Table (D&T) understanding. Even at full resolution, SPARC achieves comparable or slightly superior performance globally. These results confirm that decoupling perception from reasoning does not induce a penalty on general vision-language capabilities.

### A.13. Expanded Results Tables

We present expanded performance metrics across varying computational budgets for $V^*$, HRBench-4K, and HRBench-8K. Table 14 details the results for the SFT-based SPARC experiments. Table 15 provides the corresponding performance data for WBF, while Table 16 reports the associated crop counts. For the "thinking with images" baselines we follow the official implementation script with off-the-shelf parameters: $top\_p = 0.8$, $top\_k = 20$, $temperature = 0.7$, $repetition\_penalty = 1.0$, $presence\_penalty = 1.5$.

### A.14. Qualitative Analysis

Figure 13 presents qualitative results from the WBF experiment. The merging algorithm proves particularly effective at lower resolutions, where lower confidence levels lead the model to generate a diverse set of bounding boxes. At full

*Table 12.* Evaluation on the MME Realworld benchmark spanning Perception and Reasoning tasks across multiple domains: Autonomous Driving (Auto.), Diagrams and Tables (D&T), Monitoring (Mon.), Optical Character Recognition (OCR), and Remote Sensing (RS).

| Method | Res. | All | Perception (%) | | | | | Reasoning (%) | | | |
|---|---|---|---|---|---|---|---|---|---|---|---|
| | | | Auto. | D&T | Mon. | OCR | RS | Auto. | D&T | Mon. | OCR |
| *Qwen3-VL 4B Native* | 256 | 28.4 | 27.7 | 10.0 | 25.4 | 42.0 | 20.7 | 29.5 | 19.0 | 32.7 | 35.0 |
| | 512 | 35.8 | 33.4 | 39.0 | 25.7 | 56.0 | 30.7 | 28.5 | 33.0 | 41.3 | 53.0 |
| | Full | 50.7 | 42.9 | 87.0 | 34.2 | 91.6 | 47.3 | 30.5 | 63.0 | 44.0 | 75.0 |
| *Qwen3-VL 4B SPARC* | 256 | 32.8 | 29.1 | 19.0 | 27.3 | 56.0 | 23.3 | 28.6 | 36.0 | 33.3 | 48.0 |
| | 512 | 39.2 | 32.3 | 42.0 | 32.9 | 70.8 | 29.3 | 27.0 | 43.0 | 43.3 | 56.0 |
| | Full | 52.5 | 41.7 | 86.0 | 42.6 | 90.4 | 50.8 | 30.8 | 64.0 | 50.8 | 74.0 |
| *Qwen3-VL 8B Native* | 256 | 28.7 | 24.6 | 19.0 | 27.3 | 44.0 | 24.7 | 26.5 | 29.0 | 28.0 | 36.0 |
| | 512 | 34.1 | 27.4 | 41.0 | 28.8 | 54.8 | 28.7 | 26.3 | 33.0 | 35.3 | 55.0 |
| | Full | 50.5 | 34.6 | 86.0 | 35.4 | 93.6 | 58.0 | 30.5 | 67.0 | 42.0 | 76.0 |
| *Qwen3-VL 8B SPARC* | 256 | 26.1 | 17.1 | 18.0 | 23.8 | 48.4 | 22.7 | 20.5 | 28.0 | 30.0 | 37.0 |
| | 512 | 36.4 | 24.3 | 45.0 | 33.5 | 68.8 | 28.7 | 23.5 | 38.0 | 36.0 | 60.0 |
| | Full | 50.0 | 32.3 | 86.0 | 43.6 | 91.2 | 50.0 | 28.5 | 67.0 | 40.7 | 77.0 |

resolution, the WBF results in a deduplication of virtually the same bounding boxes. Additionally, Figures 6 through 9 provide qualitative comparisons between the "thinking with images" baseline and SPARC across various use cases.

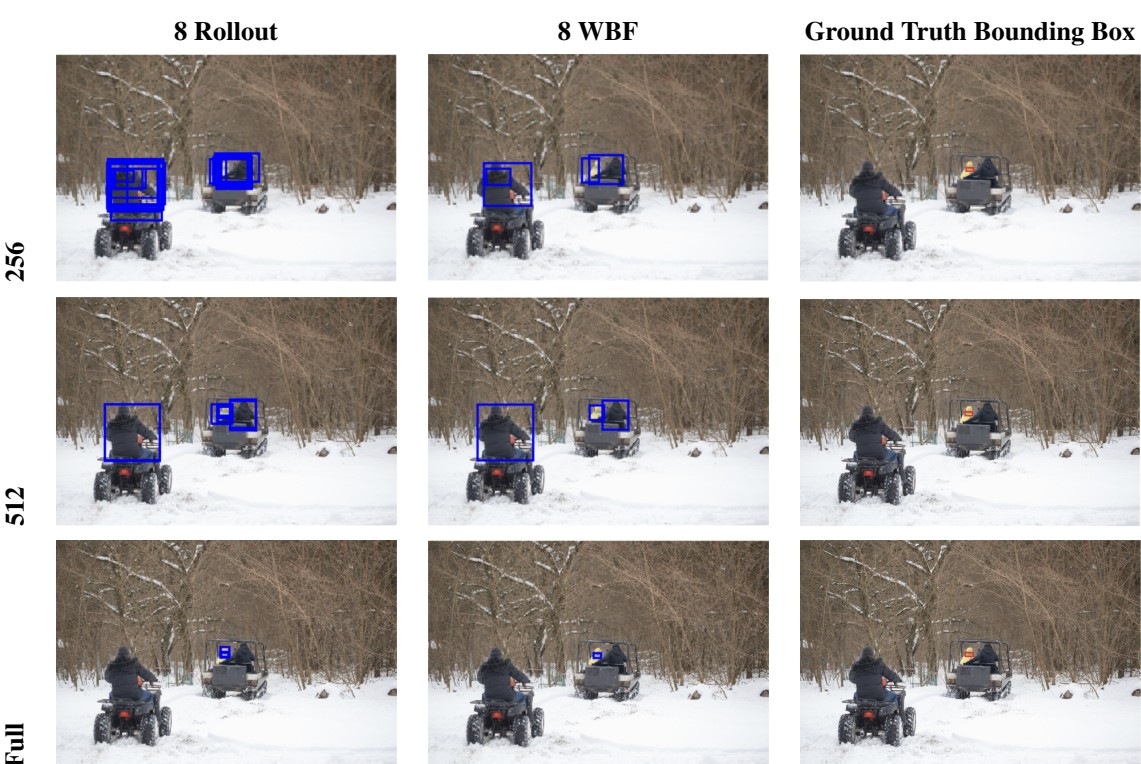

*Table 13.* Qualitative comparison across resolutions (256, 512, Full) for 8 Rollouts WBF, and Ground Truth.

*Table 14.* Expanded table of results for the SPARC SFT experiments

| Setting | V* | | | HRBench-4K | | | HRBench-8K | | | Average | | |
|---|---|---|---|---|---|---|---|---|---|---|---|---|
| | 256 | 512 | Full | 256 | 512 | Full | 256 | 512 | Full | 256 | 512 | Full |
| *Qwen3-VL 4B* | | | | | | | | | | | | |
| Native Performance | 40.3 | 47.9 | 84.5 | 43.5 | 52.8 | 68.8 | 41.4 | 45.6 | 64.6 | 41.7 | 48.8 | 72.6 |
| "Thinking w/ images" | 32.1 | 51.5 | 83.1 | 43.1 | 58.4 | **72.0** | 35.1 | 46.8 | 64.4 | 36.8 | 52.2 | 73.1 |
| SPARC | 54.6 | 66.4 | 86.1 | 53.8 | 60.0 | 69.0 | 44.8 | 55.4 | **69.4** | 51.0 | 60.6 | 74.8 |
| SPARC SFT Full | 53.4 | 66.0 | 87.8 | 52.8 | 59.4 | 70.5 | 46.4 | 50.4 | 68.4 | 50.8 | 58.6 | 75.6 |
| SPARC SFT 512 | 54.6 | 65.5 | 88.2 | 52.3 | 59.4 | 68.5 | 46.0 | 52.4 | 67.1 | 51.0 | 59.1 | 74.6 |
| SPARC SFT 256 | **55.9** | **69.3** | 91.2 | 53.9 | 63.4 | 70.6 | **45.4** | 59.4 | 68.5 | 51.7 | 64.0 | 76.8 |
| *Qwen3-VL 8B* | | | | | | | | | | | | |
| Native Performance | 38.2 | 46.2 | 87.0 | 43.6 | 55.6 | 77.0 | 42.3 | 46.3 | 73.0 | 41.4 | 49.4 | 79.0 |
| "Thinking w/ images" | 35.3 | 55.9 | 86.6 | 47.5 | 61.4 | 76.6 | 40.0 | 52.4 | 71.0 | 40.9 | 56.5 | 78.1 |
| SPARC | 50.0 | 62.6 | 88.7 | 43.6 | 55.8 | 76.9 | 42.5 | 46.1 | 73.0 | 45.4 | 54.8 | 79.5 |
| SPARC SFT Full | 52.5 | 64.3 | 91.6 | 43.6 | 55.8 | 76.8 | 42.9 | 46.4 | 73.0 | 46.3 | 55.5 | 80.4 |
| SPARC SFT 512 | 46.6 | 69.3 | **92.9** | 43.6 | 55.8 | 76.8 | 42.9 | 46.3 | 73.0 | 44.4 | 57.1 | 80.9 |
| SPARC SFT 256 | **52.9** | 69.7 | 91.2 | 59.4 | 65.6 | 79.5 | 47.0 | 57.6 | 76.4 | 53.1 | 64.3 | 82.4 |
| *Molmo2 4B* | | | | | | | | | | | | |
| Native Performance | 48.3 | 56.7 | 71.4 | **50.0** | 55.8 | 57.9 | **46.1** | **47.1** | 53.1 | 48.1 | 53.2 | 60.8 |
| SPARC | **55.0** | **70.2** | 76.5 | 47.5 | 54.1 | 57.8 | 43.5 | 46.8 | **54.5** | **48.7** | **57.0** | 62.9 |
| SPARC SFT Full | 49.6 | 60.1 | 74.8 | 47.3 | 54.1 | 57.8 | 44.0 | 44.0 | 51.6 | 46.9 | 52.7 | 61.4 |
| SPARC SFT 512 | 53.4 | 62.2 | 74.4 | 46.1 | 53.5 | 57.5 | 42.0 | 44.5 | 51.1 | 47.2 | 53.4 | 61.0 |
| SPARC SFT 256 | 53.4 | 66.8 | **79.4** | 47.4 | **56.8** | **58.5** | 43.8 | 46.9 | 53.4 | 48.2 | 56.8 | **63.8** |
| *Molmo2 8B* | | | | | | | | | | | | |
| Native Performance | 46.2 | 56.3 | 68.5 | 47.8 | 56.4 | 55.3 | 43.4 | 44.5 | 49.6 | 45.8 | 52.4 | 57.8 |
| SPARC | 52.9 | 62.6 | 70.2 | 46.5 | 56.6 | 58.1 | 42.8 | **45.6** | 49.1 | 47.4 | 55.0 | 59.1 |
| SPARC SFT Full | **53.4** | 63.0 | 67.7 | 47.4 | 53.9 | 55.9 | 42.8 | **45.6** | 49.0 | 47.8 | 54.2 | 57.5 |
| SPARC SFT 512 | 52.9 | 61.3 | 71.9 | 48.0 | 54.8 | 55.8 | 44.6 | 43.5 | **50.4** | **48.5** | 53.2 | 59.3 |
| SPARC SFT 256 | 51.7 | **66.0** | 72.3 | 48.4 | 56.3 | 57.6 | 44.4 | 45.3 | 48.9 | 48.1 | **55.8** | 59.6 |

*Table 15.* Expanded table of results for the SPARC WBF experiments

| Setting | V* | | | HRBench-4K | | | HRBench-8K | | | Average | | |
|---|---|---|---|---|---|---|---|---|---|---|---|---|
| | 256 | 512 | Full | 256 | 512 | Full | 256 | 512 | Full | 256 | 512 | Full |
| *Qwen3-VL 4B* | | | | | | | | | | | | |
| Native Performance | 40.3 | 47.9 | 84.5 | 43.5 | 52.8 | 68.8 | 41.4 | 45.6 | 64.6 | 41.7 | 48.8 | 72.6 |
| "Thinking w/ images" | 32.1 | 51.5 | 83.1 | 43.1 | 58.4 | 72.0 | 35.1 | 46.8 | 64.4 | 36.8 | 52.2 | 73.1 |
| SPARC | 54.6 | 66.4 | 86.1 | 53.8 | 60.0 | 69.0 | 44.8 | 55.4 | 69.4 | 51.0 | 60.6 | 74.8 |
| SPARC WBF 4 | 54.6 | **69.3** | **87.0** | 58.1 | 68.1 | 79.4 | 50.0 | 58.6 | **78.8** | 54.2 | 65.4 | 81.7 |
| SPARC WBF 8 | **56.6** | 68.9 | **87.0** | 60.3 | 70.8 | 80.4 | 50.1 | 61.3 | 78.6 | 55.7 | 67.0 | 82.0 |
| *Qwen3-VL 8B* | | | | | | | | | | | | |
| Native Performance | 38.2 | 46.2 | 87.0 | 43.6 | 55.6 | 77.0 | 42.3 | 46.3 | 73.0 | 41.4 | 49.4 | 79.0 |
| "Thinking w/ images" | 35.3 | 55.9 | 86.6 | 47.5 | 61.4 | 76.6 | 40.0 | 52.4 | 71.0 | 40.9 | 56.5 | 78.1 |
| SPARC | **50.0** | 62.6 | 88.7 | 43.6 | 55.8 | 76.9 | 42.5 | 46.1 | 73.0 | 45.4 | 54.8 | 79.5 |
| SPARC WBF 4 | 48.3 | 63.9 | 88.2 | 55.3 | 67.9 | **80.9** | 43.0 | 58.5 | 73.5 | 48.9 | 63.4 | 80.9 |
| SPARC WBF 8 | 49.2 | **64.7** | **89.5** | **56.0** | **68.9** | 79.6 | **44.5** | **58.6** | **74.1** | 49.9 | 64.1 | 81.1 |
| *Molmo2 4B* | | | | | | | | | | | | |
| Native Performance | 48.3 | 56.7 | 71.4 | **50.0** | **55.8** | **57.9** | **46.1** | 47.1 | 53.1 | 48.1 | 53.2 | 60.8 |
| SPARC | 55.0 | 70.2 | 76.5 | 47.5 | 54.1 | 57.8 | 43.5 | 46.8 | **54.5** | 48.7 | 57.0 | 62.9 |
| SPARC WBF 4 | 55.0 | 71.0 | 78.6 | 47.1 | 54.5 | 57.6 | 44.0 | 47.8 | 53.8 | 48.7 | 57.8 | 63.3 |
| SPARC WBF 8 | **64.7** | **72.3** | **80.7** | 48.6 | 54.4 | **57.9** | 44.5 | **48.4** | 53.9 | **52.6** | **58.3** | **64.1** |
| *Molmo2 8B* | | | | | | | | | | | | |
| Native Performance | 46.2 | 56.3 | 68.5 | 47.8 | 56.4 | 55.3 | **43.4** | 44.5 | **49.6** | 45.8 | 52.4 | 57.8 |
| SPARC | 52.9 | 62.6 | **70.2** | 46.5 | 56.6 | **58.1** | 42.8 | 45.6 | 49.1 | 47.4 | 55.0 | **59.1** |
| SPARC WBF 4 | 52.9 | 64.7 | 68.1 | 48.6 | 56.8 | 57.8 | 42.3 | 43.4 | 48.8 | 47.9 | 54.9 | 58.2 |
| SPARC WBF 8 | **53.4** | **66.4** | 68.9 | **48.8** | **57.5** | 57.3 | 41.9 | **43.8** | 48.6 | **48.0** | **55.9** | 58.3 |

*Table 16.* Expanded table of results on the number of crops for the SPARC WBF experiments

| Setting | V* | | | HRBench-4K | | | HRBench-8K | | | Average | | |
|---|---|---|---|---|---|---|---|---|---|---|---|---|
| | 256 | 512 | Full | 256 | 512 | Full | 256 | 512 | Full | 256 | 512 | Full |
| *Qwen3-VL 4B* | | | | | | | | | | | | |
| Predicted Crops | 1.84 | 1.79 | 1.67 | 1.50 | 1.58 | 1.62 | 1.42 | 1.53 | 1.63 | 1.59 | 1.63 | 1.64 |
| SPARC WBF 4 | 2.19 | 1.83 | 1.43 | 2.40 | 2.07 | 1.83 | 2.54 | 2.26 | 1.91 | 2.38 | 2.05 | 1.72 |
| SPARC WBF 8 | 3.14 | 2.20 | 1.52 | 3.27 | 2.52 | 2.06 | 3.48 | 2.90 | 2.06 | 3.30 | 2.54 | 1.88 |
| *Qwen3-VL 8B* | | | | | | | | | | | | |
| Predicted Crops | 1.08 | 1.18 | 1.30 | 1.36 | 1.45 | 1.57 | 1.26 | 1.30 | 1.50 | 1.23 | 1.31 | 1.46 |
| SPARC WBF 4 | 1.79 | 1.59 | 1.36 | 2.05 | 1.93 | 1.79 | 2.03 | 1.79 | 1.82 | 1.96 | 1.77 | 1.66 |
| SPARC WBF 8 | 2.22 | 1.90 | 1.48 | 2.65 | 2.42 | 1.98 | 2.75 | 2.25 | 1.99 | 2.54 | 2.19 | 1.82 |
| *Molmo2 4B* | | | | | | | | | | | | |
| Predicted Crops | 2.20 | 1.64 | 1.58 | 1.85 | 1.53 | 1.83 | 1.84 | 1.70 | 1.77 | 1.96 | 1.62 | 1.72 |
| SPARC WBF 4 | 2.48 | 1.97 | 1.91 | 3.46 | 2.71 | 2.78 | 4.66 | 3.60 | 3.50 | 3.53 | 2.76 | 2.73 |
| SPARC WBF 8 | 3.64 | 2.48 | 2.46 | 5.30 | 3.92 | 4.03 | 7.78 | 5.86 | 5.60 | 5.57 | 4.09 | 4.03 |
| *Molmo2 8B* | | | | | | | | | | | | |
| Predicted Crops | 1.50 | 1.47 | 1.36 | 1.59 | 1.72 | 1.65 | 1.56 | 1.69 | 1.60 | 1.55 | 1.63 | 1.54 |
| SPARC WBF 4 | 1.71 | 1.48 | 1.44 | 2.60 | 2.27 | 2.25 | 3.40 | 2.92 | 2.73 | 2.57 | 2.22 | 2.14 |
| SPARC WBF 8 | 2.09 | 1.77 | 1.66 | 3.96 | 3.27 | 3.03 | 5.24 | 4.42 | 4.06 | 3.76 | 3.15 | 2.92 |

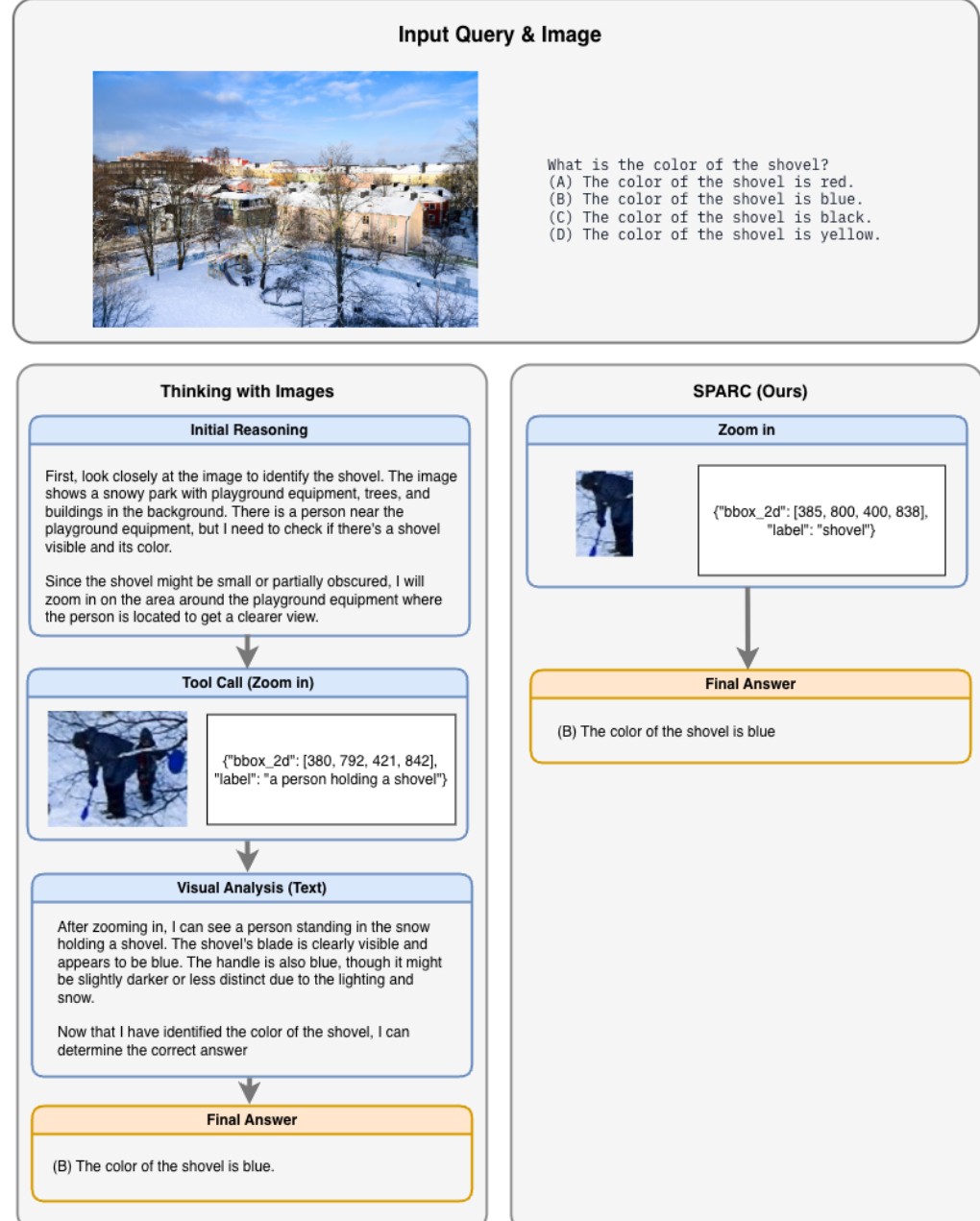

*Figure 6.* While both models answer correctly, "thinking with images" (left) relies on a dense, unstructured chain-of-thought, consuming a large token budget to plan and describe the scene. On the other hand **SPARC** (right) isolates the target object and answers instantly with significantly lower computational cost.

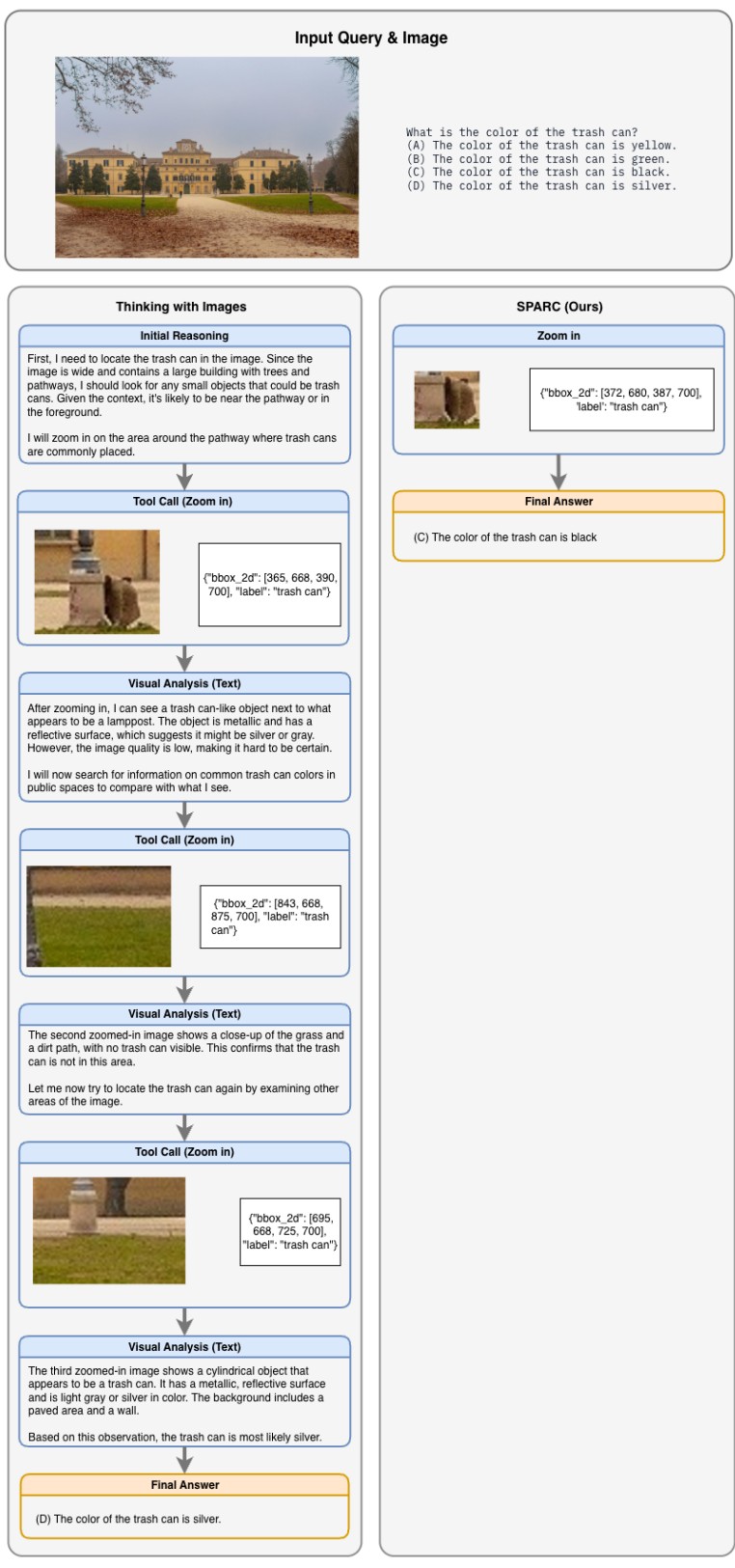

*Figure 7.* On the left, "thinking with images" initially isolates the correct crop but misinterprets the visual content due to a deceptive text description. This misalignment triggers a series of wasteful search steps, leading the model to confuse a stone lamp post base for a trash can. Consequently, it hallucinates 'metallic' and 'reflective' properties, resulting in an incorrect 'Silver' prediction. On the right, **SPARC** correctly localizes the actual black bin immediately, avoiding the reasoning trap and returning the correct answer in a single step.

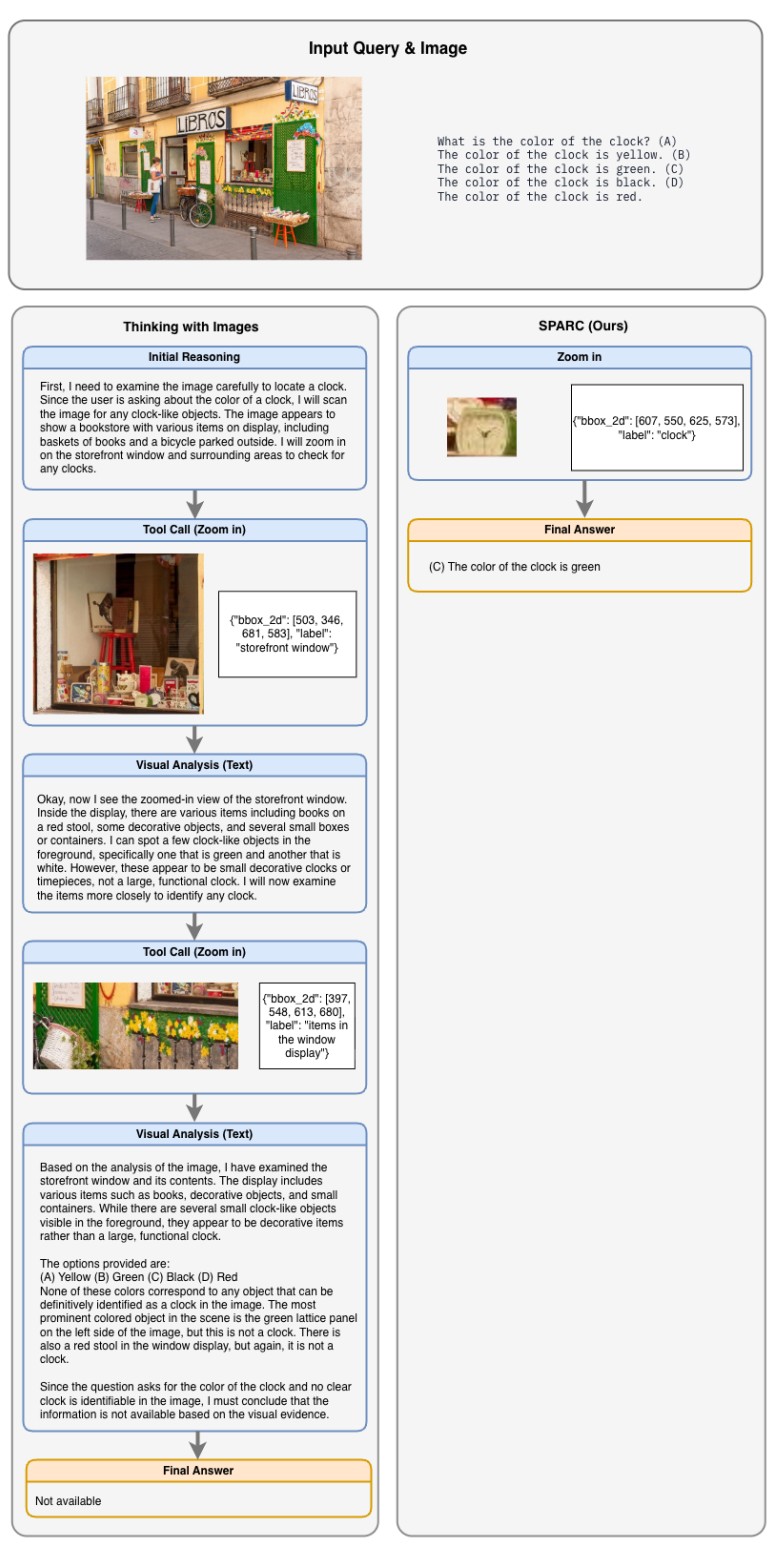

*Figure 8.* "Thinking with images" correctly spots the green clock but hallucinates that it must be 'large and functional,' causing it to discard valid visual evidence, a classic example of how reasoning errors cascade in monolithic models. On the right, **SPARC** succeeds by decoupling perception: it explicitly localizes the clock first via visual search, isolating the relevant region before reasoning begins, effectively preventing prior bias and arriving at the correct answer with significantly fewer tokens.

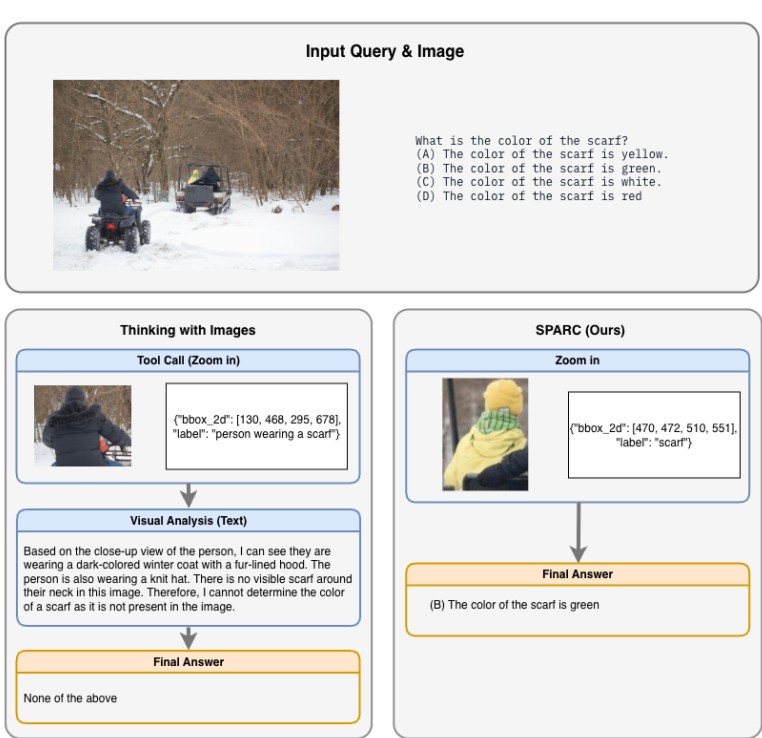

*Figure 9.* "Thinking with images" concentrates on the most prominent foreground subject, correctly reasoning that this person has no scarf, but failing to scan the background for the actual target. On the right, **SPARC** demonstrates the benefit of explicit visual search. It successfully localizes the smaller background figure wearing the green scarf.

