# OpenReview forum: "SPARC: Separating Perception And Reasoning Circuits for Test-time Scaling of VLMs"
_ICML.cc/2026/Conference — ICML 2026 regular_

### Official Review · Reviewer_gEb6 · 2026-02-14

**Soundness:** 3
**Presentation:** 2
**Significance:** 3
**Originality:** 2
**Overall Recommendation:** 4
**Confidence:** 5

**Summary:**

The paper introduces SPARC, a two-stage inference framework for Vision-Language Models (VLMs) that decouples visual perception from reasoning. The method involves an initial Implicit Relevance Detection (IRD) stage where the model outputs coordinates (boxes/points) for relevant regions, followed by a reasoning stage conditioned on these high-resolution crops. The authors claim this separation allows for independent test-time scaling (via ensemble cropping), asymmetric compute allocation, and efficient fine-tuning. Experiments on V*, HRBench, and XLRS show performance gains over native and "thinking with images" baselines, particularly in low-resolution and out-of-distribution settings.

**Compliance With Llm Reviewing Policy:**

Affirmed.

**Final Justification:**

The authors addressed what I pointed out well.

**Key Questions For Authors:**

* In the SFT experiments, did you attempt to fine-tune the baseline models on the teacher's (235B) full textual reasoning traces? This is necessary to prove that the *separation* of circuits is better than standard distillation.
* Can you provide wall-clock latency comparisons between SPARC (with  rollouts + WBF) and the single-pass "Thinking with Images" baseline?
* How does the method handle queries where the answer depends on the absence of an object or global scene properties (e.g., "Is the room crowded?") where no specific crop is "relevant"?

**Limitations:**

- Requires native or robust coordinate/point generation, limiting applicability to a subset of VLMs.
- May degrade on tasks needing global context or multi-hop visual relations where cropping introduces missing evidence or distractors.
- Efficiency claims are not fully grounded in end-to-end serving cost metrics (latency/memory/throughput).
- Training improvements rely on synthetic trace generation + filtering, and performance may track teacher quality rather than method robustness

**Strengths And Weaknesses:**

Strengths:
1. Modular Design: The explicit separation of "where to look" (IRD) and "what to think" aligns well with principles of visual context engineering and allows for interpretable intermediate outputs.
2. Effective for Low-Res/OOD: The method demonstrates strong empirical results on remote sensing data (XLRS), suggesting that "cropping to relevance" is a viable strategy for handling high-resolution data under strict token budgets.
3. Practical Scaling Mechanism: The application of self-consistency specifically to the coordinate generation step (via Weighted Box Fusion) is a clever, compute-efficient way to improve reliability without repeating the full reasoning chain.

Weaknesses:
1. Conflation of Distillation and Architecture (Major):
The fine-tuning experiments (Section 6) rely on synthetic data generated by a significantly larger teacher model (Qwen3-VL-235B) to train the 4B student. It is difficult to disentangle how much of the performance gain is attributable to the SPARC architecture versus simply distilling the superior reasoning traces of a 235B model into the student. The paper should ideally compare against a baseline where the 4B model is simply fine-tuned on the *reasoning* traces of the teacher, not just the *cropping* traces, to isolate the architectural benefit.
2. Efficiency Metrics are Under-Specified:
The paper heavily emphasizes efficiency based on "token count" (e.g., "200x lower token budget" ). However, for real-world deployment, token count is a proxy, not a direct measure of cost. The evaluation lacks wall-clock latency, Time-To-First-Token (TTFT), and system overhead measurements. The proposed pipeline requires multiple model invocations (IRD + Reasoning) and CPU-bound post-processing (Weighted Box Fusion), which may introduce latency penalties that negate the token savings.
3. Incremental Novelty & Framing:
While the packaging is cohesive, the core mechanism—prompting for coordinates and then reprompting with crops—closely resembles existing "Ground-then-Reason" or "Crop-Zoom-Reason" pipelines. The framing relies heavily on neuroscience analogies (What/Where pathways) to elevate what is essentially a prompt-engineering and ensemble strategy.
4. Baseline Fairness:
The comparison to the "thinking with images" baseline uses "off-the-shelf generation parameters". Multi-modal Chain-of-Thought typically requires careful tuning of temperature and sampling to be effective. If the baseline was run with greedy decoding (as SPARC was ), it may have artificially handicapped the baseline's ability to explore reasoning paths, creating a strawman comparison.
5. Ill-Posed Objective & Context Loss:
As the authors admit, "relevance" is ill-posed. A strict cropping approach risks stripping essential global context or relational information (e.g., an object's size relative to the room). The evaluation does not sufficiently analyze failure modes where tight crops lead to "tunnel vision," causing the model to miss negative constraints or global scene semantics.

---

> ### Author Rebuttal · Authors · 2026-03-31
>
> We thank the reviewer for recognizing SPARC's modularity, OOD/low-res effectiveness, and WBF compute efficiency.
>
> ### Distillation SFT Baseline
>
> A central question is whether SPARC's gains are attributable to its architecture or merely to distillation of the 235B teacher's reasoning traces. To isolate the architectural contribution, we fine-tuned the 4B and 8B models directly on the teacher's full multimodal reasoning traces.
>
> |**Setting**|**256**|**512**|**Full**|
> |---|---|---|---|
> |**4B Models**||||
> |TwI (Zero-shot)|36.8|52.2|73.2|
> |Distillation SFT|37.7|50.2|66.6|
> |**SPARC SFT**|**51.7**|**64.0**|**76.8**|
> |**8B Models**||||
> |TwI (Zero-shot)|40.9|56.5|78.1|
> |Distillation SFT|42.7|54.9|72.1|
> |**SPARC SFT**|**53.1**|**64.3**|**82.4**|
>
> **Findings:** Distilling the full reasoning traces degrades overall performance relative to the zero-shot "Thinking with Images" (TwI) baseline. Given the small and synthetic dataset, standard end-to-end distillation induces catastrophic forgetting, undermining the student's general reasoning priors (often resulting in degenerate repetitive tool calls). SPARC’s performance gains are therefore of architectural origin. Isolating the perception module enables aggressive optimization of localization on noisy data without compromising reasoning capabilities. We attribute SPARC SFT's resistance to catastrophic forgetting to the IRD objective, a well-scoped task that does not disrupt the extensive prior SFT+RL alignment of these models.
>
> ### Baseline Fairness and Decoding Parameters
>
> We apologize for the lack of clarity. By "off-the-shelf generation parameters," we refer to the official parameters specified in the Qwen3 technical report. Specifically, we used the recommended temperature (1 or 0.7, depending on the model size), not greedy decoding. We will explicitly state the sampling configurations for TwI in the revised manuscript.
>
> ### Global Context, Context Loss, and "Tunnel Vision"
>
> The reviewer correctly observes that global or "absence" queries cannot be solved by tight crops alone. Importantly, however, SPARC does not discard the global image. The reasoning module is conditioned on both the high-resolution crop and the original full image, thereby retaining access to global context when required. To empirically assess that SPARC avoids "tunnel vision", we evaluated it on the MME benchmark (detailed in our response to Reviewer W8Kx, section Benchmark Coverage), where it consistently matches or slightly outperforms the native model on general reasoning tasks.
>
> ### Efficiency Metrics (TTFT)
>
> We rely on token count as our primary efficiency metric because it is independent of specific hardware or model implementations. This makes it a reliable and reproducible metric for assessing computational overhead. In addition, it has the practical aspect of serving as the standard pricing metric for commercial inference APIs, which makes it the main concern for practitioners.
> Nevertheless, we understand and share the reviewer's interest in other metrics such as latency. To address this, we conducted a server-side Time to First Token (TTFT) analysis comparing SPARC and TwI. While we provide these numbers and will be happy to include them in the revisions, we'd like to caution that TTFT is a brittle, implementation-dependent and hardware-bound metric that can depend on server load and framework optimizations.
>
> |**TTFT**|**SPARC**|**Twi**|
> |---|---|---|
> |**V***|||
> |256|0.2006|0.2846|
> |Full|2.9674|3.0494|
> |**_XLRS_**|||
> |256|0.0598|0.0744|
> |Full|13.1873|13.5052|
>
> Across all settings SPARC achieves lower latency than TwI. Furthermore, given that SPARC at 256 resolution outperforms TwI in overall performance (IoU) at full resolution, the proposed pipeline provides higher accuracy while being over 200x faster in terms of TTFT.
>
> ### Novelty and Limitations
>
> Regarding novelty, SPARC’s core strength lies precisely in its practical simplicity, offering a robust and scalable architectural solution to a documented crisis in VLM reliability. Recent literature reveals that VLMs face a "perceptual bottleneck" collapse into biased "mirage reasoning," and forcing them into extended multimodal reasoning chains only causes cascading hallucinations over invisible features. By strictly decoupling perception from reasoning, SPARC actively prevents this context degradation, while significantly differentiating from  standard "ground-then-reason" pipelines, as it uses IRD instead of external grounding and accommodates principled test-time scaling via WBF. Please refer to our response to Reviewer Q5Qk (section Relationship with Existing Literature) for further details on how our top-down approach differentiates from standard divide-and-conquer baselines.
>
> We acknowledge that SPARC requires a model capable of coordinate generation. However, this limitation applies equally to all "Thinking with Images" and agentic baselines. We will address this constraint in the conclusion section of the revised manuscript.

---

> > ### Author Rebuttal · Reviewer_gEb6 · 2026-04-01
> >
> > Thank you again for the detailed rebuttal. I have a follow-up question regarding efficiency and system-level cost.
> >
> > While the provided TTFT numbers are helpful, the SPARC pipeline involves multiple stages (IRD prediction, potential rollouts, weighted box fusion, and a second-stage reasoning pass). Could you clarify the full end-to-end latency and compute overhead compared to the TwI baseline under the same deployment setting? In particular, it would be helpful to understand how the additional model invocations and CPU-side post-processing impact throughput and total inference time, beyond token-level metrics.
> >
> > Additionally, could you comment on how SPARC performs on queries that depend on global scene understanding or the absence of objects (e.g., “Is the room empty?” or relational comparisons across distant regions)? A targeted analysis or example would help clarify whether the cropping mechanism introduces any systematic failure modes in such cases.

---

> > > ### Author Response · Authors · 2026-04-01
> > >
> > > We sincerely thank the reviewer for engaging with our rebuttal and for the opportunity to clarify the missing points.
> > > ### End-to-End Latency and Compute Overhead
> > > To address the concern regarding throughput and total inference time, we present the full End-to-End latency comparisons below:
> > >
> > > | **E2E Latency [s]** | **SPARC** | **TwI** |
> > > | ------------------- | ------------ | ---------- |
> > > | **V***            |              |            |
> > > | 256                 | 0.663        | 1.861      |
> > > | Full                | 4.767        | 6.116      |
> > > | **_XLRS_**          |              |            |
> > > | 256                 | 0.332        | 2.138      |
> > > | Full                | 13.573       | 15.857     |
> > >
> > > The data demonstrates that SPARC is consistently and significantly faster than the TwI baseline. Regarding the CPU-side post-processing: this overhead is asynchronous and computationally negligible compared to the GPU forward passes. In any case, both SPARC and TwI are multi-turn pipelines that incur this exact type of CPU intervention to parse intermediate responses, crop the original image, and submit the subsequent request. In fact, because TwI relies on an entangled reasoning process, it frequently triggers sequential, redundant tool calls. Consequently, TwI actually incurs a higher CPU penalty than our structured pipeline. In our profiling, the average overall CPU time for TwI was 5.38ms, compared to just 3.97ms for SPARC.
> > >
> > > ### Global Scene Understanding and Relational Queries
> > >
> > > To address the reviewer's concern, we conducted a targeted analysis on two new dataset subsets specifically designed to stress-test these exact scenarios:
> > > - V* Far Objects: A subset of the V* benchmark strictly filtered for spatial relationship queries where the two target objects are separated by at least half the image smallest size (testing relational comparisons across distant regions, 16 QA pairs).
> > > - GQA Global: A subset of the GQA test set (157 QA pairs) explicitly filtered using the "global" question tag, which includes questions requiring whole-scene understanding.
> > >
> > > | **Accuracy**       | **Resolution** | 256      | Full     |
> > > | ------------------ | -------------- | -------- | -------- |
> > > | **V * Far Objects** | Native 4B      | 62.5     | 68.8     |
> > > |                    | SPARC 4B       | **68.8** | **75.0** |
> > > | **GQA Global**     | Native 4B      | 19.1     | 24.8     |
> > > |                    | SPARC 4B       | **21.7** | **26.8** |
> > >
> > > As the results demonstrate, SPARC does not suffer from a systematic failure mode or context collapse on these queries; in fact, it outperforms the native baseline across both resolutions. This robustness is fundamentally tied to our architectural design: during the reasoning stage, SPARC does not discard the original image. The model is conditioned on both the localised high-resolution crop(s) and the original global image. If a query requires recognizing an empty room, the model naturally leverages the global image. If it requires comparing distant objects, the IRD stage either provides crops for both regions or the reasoning module simply falls back to the global view. Because the global context is never lost, SPARC safely handles scene-level semantics without penalty.

---

### Official Review · Reviewer_W8Kx · 2026-03-08

**Soundness:** 2
**Presentation:** 3
**Significance:** 3
**Originality:** 3
**Overall Recommendation:** 4
**Confidence:** 4

**Summary:**

SPARC separates visual perception from reasoning in VLMs: it first locates the image regions most relevant to a question, then performs reasoning only on those selected regions. This simple two-stage design makes test-time scaling more efficient and robust, especially for high-resolution, small-object, and out-of-distribution visual reasoning tasks.

**Compliance With Llm Reviewing Policy:**

Affirmed.

**Final Justification:**

My concerns regarding scalability have been satisfactorily addressed, and the additional experiments—particularly those involving the 235B-scale model—are convincing and provide strong evidence for the effectiveness and consistency of the proposed method; therefore, I am raising my score to 4.

**Key Questions For Authors:**

The method mainly benefits perception-heavy tasks and may provide limited gains for reasoning-dominated problems.  Will performance decrease on other benchmarks after incorporating this method design?

**Limitations:**

No limitation discussed.

**Strengths And Weaknesses:**

Strengths:

1.The paper introduces a simple yet effective framework that separates visual perception from reasoning in VLMs.

2.The two-stage pipeline (visual search → reasoning) can be integrated into existing VLMs with minimal architectural changes.

3.Strong empirical results: Experiments on multiple benchmarks show consistent improvements over strong baselines.

Weaknesses:

1.Limited conceptual novelty: The idea of grounding or locating relevant regions before reasoning is related to existing visual grounding and retrieval-based approaches.

2.Training data for region annotations is generated by one model, potentially inheriting the bias.

3.The benchmark coverage is relatively narrow, with most results focusing on high-resolution, perception-heavy settings. It remains unclear whether SPARC yields consistent gains on OCR and spatial relationship tasks, since the paper does not provide sufficiently detailed evaluations on these categories.

---

> ### Author Rebuttal · Authors · 2026-03-31
>
> We sincerely thank the reviewer for their constructive feedback and for recognizing the simplicity, effectiveness, and minimal architectural friction of the SPARC framework. We address your specific concerns below.
>
> ### Conceptual Novelty
>
> We agree that locating regions before reasoning shares DNA with classical visual grounding and retrieval approaches. However, SPARC’s novelty lies in adapting this intuition as a strict, top-down architectural intervention to solve a newly documented crisis in frontier Vision-Language Models and provide insights on the perception-reasoning trade-off to the community.
>
> Recent literature reveals a critical vulnerability: current VLMs frequently fail to extract fine-grained visual details [1]. When faced with this "perceptual bottleneck," models do not fail gracefully. Instead, they fall back on memorized biases rather than genuine visual analysis [2], a collapse into what is termed "mirage reasoning" [3]. As demonstrated in [4], forcing models into extended reasoning chains (the standard "thinking with images" paradigm) actually impairs perceptual grounding, causing them to hallucinate complex reasoning steps over invisible features.
>
> In this context, SPARC is not merely a traditional retrieval method. By strictly isolating perception from reasoning, SPARC actively prevents the cascading hallucinations and context degradation from which models operating on entangled prompts suffer when pushed to their perceptual limits. Please refer to our response to Reviewer Q5Qk section Relationship with Existing Literature, for more details on how we differentiate with respect to divide and conquer baselines.
>
> ### Training Data Bias
>
> The reviewer points out that our synthetic IRD training data is harvested from teacher traces and inherently contains noise and bias.
>
> We argue that, rather than being a weakness, this perfectly illustrates the core strength of the SPARC architecture. We deliberately used this cheap, synthetic dataset because high-quality, large-scale localization labels are not readily available. If one attempts standard end-to-end (E2E) fine-tuning on this exact dataset, the model suffers catastrophic forgetting which destroys its carefully aligned reasoning priors (an experiment we ran during this rebuttal phase, which showed E2E SFT performing worse than the zero-shot baseline, see reviewer TNvZ and gEb6).
>
> However, because SPARC strictly isolates the perception module, we can heavily optimize IRD using this imperfect, biased data to improve spatial localization without corrupting the model's reasoning capabilities. SPARC effectively pushes performance using data that actively breaks traditional E2E paradigms.
>
> ### Benchmark Coverage
> To confirm that SPARC does not degrade reasoning-dominated tasks, we provide the following clarifications and new data:
>
> - Existing Benchmarks: Our current benchmarks already cover the requested categories. V* includes extensive spatial relationship queries, and HRBench features OCR-heavy content. SPARC consistently improves performance across both.
>
> - New Evaluation (MME) [5]: We extended our evaluation to the MME Realworld benchmark, which heavily focuses on general reasoning and cognition and tabular data. Across the suite, SPARC maintains comparable (and occasionally improved) performance against native models.
>
> |**Model Setting**|**Res.**|**All**|**Perc. Auto.**|**Perc. D&T**|**Perc. Mon.**|**Perc. OCR**|**Perc. RS**|**Reas. Auto.**|**Reas. D&T**|**Reas. Mon.**|**Reas. OCR**|
> |---|---|---|---|---|---|---|---|---|---|---|---|
> |**Qwen 4B Native**|256|28.40|27.71|10.00|25.39|42.00|20.67|29.50|19.00|32.67|35.00|
> ||512|35.75|33.43|39.00|25.71|56.00|30.67|28.50|33.00|41.33|53.00|
> ||Full|50.65|42.86|87.00|34.17|91.60|47.33|30.50|63.00|44.00|75.00|
> |**Qwen 4B SPARC**|256|32.83|29.14|19.00|27.27|56.00|23.33|28.25|36.00|33.33|48.00|
> ||512|39.24|32.29|42.00|32.92|70.80|29.33|27.00|43.00|43.33|56.00|
> ||Full|52.48|41.71|86.00|42.63|90.40|50.67|30.75|64.00|50.67|74.00|
> |**Qwen 8B Native**|256|28.71|24.57|19.00|27.27|44.00|24.67|26.50|29.00|28.00|36.00|
> ||512|34.10|27.43|41.00|28.84|54.80|28.67|26.25|33.00|35.33|55.00|
> ||Full|50.50|34.57|86.00|35.42|93.60|58.00|30.50|67.00|42.00|76.00|
> |**Qwen 8B SPARC**|256|26.11|17.14|18.00|23.82|48.40|22.67|20.50|28.00|30.00|37.00|
> ||512|36.37|24.29|45.00|33.54|68.80|28.67|23.50|38.00|36.00|60.00|
> ||Full|50.03|32.29|86.00|43.57|91.20|50.00|28.50|67.00|40.67|77.00|
>
> _Abbreviations: Perc. = Perception, Reas. = Reasoning, Auto. = Autonomous, D&T = Diagram and Table, Mon. = Monitoring, RS = Remote Sensing_
>
> [1] Vision language models are blind: Failing to translate detailed visual features into words
>
> [2] Vision Language Models are Biased
>
> [3] Mirage: The Illusion of Visual Understanding
>
> [4] More Thought, Less Accuracy? On the Dual Nature of Reasoning in Vision-Language Models
>
> [5] MME-RealWorld: Could Your Multimodal LLM Challenge High-Resolution Real-World Scenarios that are Difficult for Humans?

---

> > ### Author Rebuttal · Reviewer_W8Kx · 2026-04-03
> >
> > Thank you for the detailed response and additional experiments, which help clarify several of my concerns.
> >
> > I have a follow-up question regarding scalability. The improvements appear much more significant for smaller models (e.g., 4B), while for larger models (e.g., 8B) the gains are limited and sometimes even negative. This suggests that the benefit of SPARC may diminish as the base model’s perceptual capability improves.
> >
> > Could the authors comment on how SPARC scales with model size, and whether its advantages persist for larger or stronger models?

---

> > > ### Author Response · Authors · 2026-04-03
> > >
> > > We are pleased to hear that our previous response successfully addressed most of your concerns.
> > >
> > > Regarding the scalability point, we evaluated the framework on the largest available model of the Qwen family (Qwen3VL-235B-A22B) on the V* dataset.
> > >
> > > | **Setting**          | **256**  | **512**  | **Full** |
> > > | -------------------- | -------- | -------- | -------- |
> > > | Native Baseline      | 47.5     | 52.1     | 89.5     |
> > > | Thinking with Images | 47.5     | 59.2     | 91.1     |
> > > | SPARC                | **53.0** | **65.1** | **91.6** |
> > >
> > > These results demonstrate that SPARC continues to yield consistent improvements across all resolutions, even at the 235B parameter scale, strictly outperforming the "Thinking with Images" baseline. While the relative performance delta at full resolution is narrower compared to our smaller models, this is primarily due to a ceiling effect: the 235B model's native performance (89.5%) is already near the dataset's saturation point. Despite this, SPARC still provides a definitive enhancement, pushing the absolute peak to 91.6%.
> > >
> > > Overall, consistent performance improvements have been shown across different base models (Qwen3-VL, Molmo2) and varying model sizes (4B, 8B, 235B), demonstrating that SPARC is a robust, model-agnostic architectural enhancement that scales effectively and reliably maximizes visual reasoning capabilities from the compact to the frontier scale.

---

### Official Review · Reviewer_Q5QK · 2026-03-09

**Soundness:** 3
**Presentation:** 3
**Significance:** 4
**Originality:** 2
**Overall Recommendation:** 4
**Confidence:** 4

**Summary:**

This paper introduces SPARC, a two-stage inference framework for vision-language models that separates perception from reasoning to enable more reliable and efficient test-time scaling. Stage 1 performs Implicit Relevance Detection (IRD), producing crops (boxes/points) relevant to a question; Stage 2 answers using those high-resolution crops. The method supports asymmetric scaling via perception-only self-consistency (multiple IRD rollouts merged by Weighted Box Fusion) and optional LoRA fine-tuning only for IRD, improving localization without harming reasoning. Experiments on high-resolution VQA and OOD remote sensing show consistent gains over native prompting and “thinking with images”.

**Compliance With Llm Reviewing Policy:**

Affirmed.

**Final Justification:**

The rebuttal directly answers the main concerns with new matched evidence against a bottom-up divide-and-conquer variant, concrete WBF diagnostics showing saturation rather than harmful distraction, and a convincing explanation that SPARC’s decoupled design makes noisy IRD supervision acceptable without damaging reasoning. For an stronger final version, the authors should state this more explicitly in the paper by foregrounding the efficiency–accuracy tradeoff versus bottom-up methods and briefly noting that TTFT/token-cost evidence is sufficient for rebuttal scope while fuller latency and an explicit discussion of related works should be added in the paper.

**Key Questions For Authors:**

1. Runtime and compute: Can you report end-to-end latency, memory, and total visual-token counts for SPARC vs “thinking with images” across resolutions (256/512/full), including WBF-N settings?
2. IRD quality diagnostics: What is the quantitative IRD localization quality (e.g., overlap/recall vs annotated regions) and how does it correlate with final answer accuracy across datasets, especially OOD XLRS?
3. Robustness and failure modes: When does adding more crops (higher recall) start to hurt due to context distraction, and how often does WBF merge away a necessary distinct region? Can you show controlled ablations on crop count/noise?
4. Supervision bias in IRD fine-tuning: Since IRD training traces are filtered by answer correctness and teacher behavior, how sensitive are gains to the choice of teacher, filtering threshold, and domain shift between DeepEyes-derived supervision and evaluation benchmarks?

**Limitations:**

Please see the above weakness. I will increase the scores if the authors could resolve my concerns.

**Strengths And Weaknesses:**

Strength:
1. Motivation is strong and timely: highlights brittleness of entangled multimodal CoT where small perceptual errors can cascade, and argues for structured context engineering by separating “find” from “answer.”
2. Method is simple, modular, and practical: a two-step prompting protocol that works across different grounding modalities (Qwen3-VL boxes, Molmo2 points) and enables independent optimization of perception.
3. Clear empirical improvements: SPARC improves ID averages vs native and vs “thinking with images” in several settings (Table 1), and perception-only WBF scaling yields large gains (e.g., Qwen3VL-4B Full: 74.8 → 82.0 with WBF-8; Table 2).

Weaknesses:
1. Comparative baseline coverage is incomplete: the strongest comparison is against “thinking with images,” but it is unclear how SPARC compares to other structured high-resolution perception frameworks (e.g., divide-and-conquer perception pipelines) under fully matched settings.
2. Some reported baseline behavior raises reproducibility concerns: the paper notes cases where “thinking with images” underperforms even the native model and mentions reproducibility issues, but does not fully diagnose whether this is due to decoding settings, tool-call policy, or implementation details.
3. IRD supervision construction may bias results: the IRD training data is filtered by downstream answer correctness and harvested from strong teacher traces (e.g., Qwen3-VL-235B), which may select “easy-to-explain” cases and entangle teacher/tool conventions with supervision.
4. Efficiency claims would benefit from more concrete numbers: the paper discusses token reductions and KV-cache reuse and gives a striking remote-sensing token fraction claim, but lacks systematic latency/memory measurements across models/resolutions and comparisons to alternative “crop then answer” pipelines.
5. Related work discussion: The relation between perception and reasoning is an important yet underexplored direction. This paper has proposed a sound method to maximize their synergy, however, there may be a trade-off between the two as mentioned in some works like [1]. The authors are encouraged to give some insights about the relation and differences between these works.

[1] More Thought, Less Accuracy? On the Dual Nature of Reasoning in Vision-Language Models, ICLR 2026

---

> ### Author Rebuttal · Authors · 2026-03-31
>
> ### Relationship with Existing Literature
>
> We thank the reviewer for highlighting [1]. Their finding (extended reasoning impairs perceptual grounding) perfectly aligns with SPARC. By explicitly decoupling perception and reasoning, we prevent the context degradation induced by long reasoning chains. This trade-off echoes broader concerns: VLMs often miss fine visual details [2], collapsing into bias-driven "mirage reasoning" [3,4] when facing perceptual bottlenecks. SPARC’s architecture is specifically designed to bypass it.
>
> Unlike previous methods that address visual research and reasoning by multi-turn grounding and reasoning [5-6] or brute-force, bottom-up grid slicing [7], SPARC is an efficient top-down approach, querying the model to locate content-related crops before reasoning. However, these paradigms are synergistic. We implemented a Bottom-Up SPARC: partitioning images into 3×3 grids, running independent IRD rollouts per tile, and aggregating localized boxes via weighted box fusion (WBF).
>
> |**Method**|**Average (256)**|**Average (512)**|**Average (Full)**|**Crops Number (256)**|**Crops Number (512)**|**Crops Number (Full)**|
> |---|---|---|---|---|---|---|
> |**_Qwen3VL 4B_**|||||||
> |SPARC|51.0|60.6|74.8|1.59|1.63|1.64|
> |SPARC WBF 8|55.7|67.0|**82.0**|3.30|2.54|1.88|
> |Bottom-up SPARC|**61.0**|**68.0**|81.3|5.51|5.60|5.92|
> |**_Qwen3VL 8B_**|||||||
> |SPARC|45.4|54.8|79.5|1.23|1.31|1.46|
> |SPARC WBF 8|49.9|64.1|81.1|2.54|2.19|1.82|
> |Bottom-up SPARC|**57.7**|**65.3**|**81.7**|7.33|8.21|9.28|
> |**_Molmo2 4B_**|||||||
> |SPARC|48.7|57.0|62.9|1.96|1.62|1.72|
> |SPARC WBF 8|52.6|58.3|64.1|5.57|4.09|4.03|
> |Bottom-up SPARC|**62.3**|**64.7**|**69.3**|11.60|11.05|9.84|
> |**_Molmo2 8B_**|||||||
> |SPARC|47.4|55.0|59.1|1.55|1.63|1.54|
> |SPARC WBF 8|48.0|55.9|58.3|3.76|3.15|2.92|
> |Bottom-up SPARC|**62.3**|**64.7**|**69.3**|11.60|11.05|9.84|
>
> Results confirm these structural trade-offs. Bottom-up grid processing sharply improves low-resolution performance (Molmo2 4B: 48.7 -> 62.3 at 256) by capturing fine details, but requires drastically higher crop counts (~11 vs. ~1.9). SPARC remains more efficient, achieving comparable or superior peak accuracy at high resolutions (Qwen 4B Full: 82.0 vs 81.3).
>
> [1] More Thought, Less Accuracy? On the Dual Nature of Reasoning in Vision-Language Models
>
> [2] Vision language models are blind: Failing to translate detailed visual features into words
>
> [3] Vision Language Models are Biased
>
> [4] Mirage: The Illusion of Visual Understanding
>
> [5] ZoomEye: Enhancing Multimodal LLMs with Human-Like Zooming
>
> Capabilities through Tree-Based Image Exploration
>
> [6] Grounded Reinforcement Learning for Visual Reasoning
>
> [7] Divide, Conquer and Combine: A Training-Free Framework for High-Resolution
>
> ### Efficiency Metrics
> While we prioritize token counts to reliably measure inference costs, our TTFT analysis confirms SPARC achieves lower latency. Please see our response to Reviewer TNvZ for detailed metrics.
>
> ### Bias in Training Dataset
> The noted noise in our synthetic IRD data actually highlights SPARC's architectural strength. Standard end-to-end (E2E) fine-tuning on this imperfect dataset causes catastrophic forgetting, destroying reasoning priors (see response to TNvZ, section Distilled SFT Baseline). Because SPARC strictly isolates perception, we can optimize spatial localization using this cheap data without corrupting aligned reasoning capabilities. SPARC thrives on data that breaks traditional E2E models.
>
> ### WBF Diagnostics
> To analyze failure modes and WBF behavior, we conducted a diagnostic ablation on V* (we cannot do that over XLRS due to missing OD labels), comparing raw IRD rollouts versus WBF deduplication up to 64 rollouts.
>
>
> |**Rollouts**|**4**|**8**|**16**|**32**|**64**|
> |---|---|---|---|---|---|
> |**_Rollout Metrics_**||||||
> |Area|0.89%|0.94%|1.17%|1.30%|1.78%|
> |Intersection|0.19%|0.19%|0.19%|0.19%|0.20%|
> |Accuracy|88.2|87.8|88.2|89.0|88.0|
> |Crops Number|5.4|10.5|21.3|42.6|85.4|
> |**_WBF Metrics_**||||||
> |Area|0.79%|0.80%|1.00%|1.09%|1.47%|
> |Intersection|0.18%|0.18%|0.18%|0.18%|0.18%|
> |Accuracy|88.2|89.5|88.7|89.1|88.6|
> |Crop Number|1.57|1.68|1.93|2.09|2.44|
> |**_Overall Metrics_**||||||
> |GT Area|0.20%|0.20%|0.20%|0.20%|0.20%|
> |Coverage loss|3.88%|5.46%|6.65%|7.00%|6.72%|
>
> Key Takeaways:
> - The target objects in V* are extremely small (GT Area is ~0.20% of the image). Even with 64 naive rollouts (85 total crops), the union area remains tight (1.78%). This proves IRD almost never points to random, far-off regions in this dataset.
> - WBF peaks around 8 rollouts because the intersection with the Ground Truth is already near-perfect (0.19% out of 0.20%). Adding more crops beyond 8 doesn't hurt due to context distraction; it simply offers no new visual information, resulting in stable but saturated accuracy.
> - WBF successfully compresses 85 raw crops (at 64 rollouts) down to just ~2.4 highly relevant crops, retaining max accuracy while maintaining strict context efficiency.

---

> > ### Author Rebuttal · Reviewer_Q5QK · 2026-04-04
> >
> > The rebuttal directly answers the main concerns with new matched evidence against a bottom-up divide-and-conquer variant, concrete WBF diagnostics showing saturation rather than harmful distraction, and a convincing explanation that SPARC’s decoupled design makes noisy IRD supervision acceptable without damaging reasoning. For an stronger final version, the authors should state this more explicitly in the paper by foregrounding the efficiency–accuracy tradeoff versus bottom-up methods and briefly noting that TTFT/token-cost evidence is sufficient for rebuttal scope while fuller latency and an explicit discussion of related works should be added in the paper.

---

> > > ### Author Response · Authors · 2026-04-04
> > >
> > > We thank the reviewer for the constructive feedback and are pleased that our rebuttal resolved the primary concerns. As recommended, we will fully incorporate the new material into the final manuscript. This includes explicitly highlighting the efficiency-accuracy tradeoff against bottom-up methods, further expanding on latency and token-cost analyses, and providing a comprehensive discussion of the related literature. We appreciate the reviewer's insights, which have significantly strengthened the framing of our work.

---

### Official Review · Reviewer_TNvZ · 2026-03-12

**Soundness:** 3
**Presentation:** 3
**Significance:** 2
**Originality:** 2
**Overall Recommendation:** 4
**Confidence:** 3

**Summary:**

This paper studies test-time scaling for vision-language models (VLMs). The paper proposes SPARC, a framework that separates visual perception and reasoning during inference. Instead of interleaving reasoning and visual actions (“thinking with images”), the framework first performs Implicit Relevance Detection (IRD) to locate question-relevant image regions and then performs reasoning conditioned on the selected crops. The goal is to reduce long multimodal chains of thought and improve efficiency by organizing visual context more explicitly. The framework is evaluated on several visual reasoning benchmarks including V*, HRBench, and XLRS using models such as Qwen3-VL and Molmo2. The experiments show that the proposed pipeline improves accuracy over native VLM inference and some grounded reasoning baselines while also reducing the visual token budget in high-resolution scenarios.

**Compliance With Llm Reviewing Policy:**

Affirmed.

**Final Justification:**

Overall, it is a complete and well-organized paper, and the empirical results are convincing. The only remaining concerns are its practical usefulness and its long-term significance.

**Key Questions For Authors:**

How does the framework perform when applied to larger models or more complex multimodal tasks, such as multi-image or video reasoning? These scenarios seem to be more perception-heavy, and require more perception-augmented strategies.

**Limitations:**

yes

**Strengths And Weaknesses:**

### Strengths
1. The proposed pipeline is simple and easy to implement. The separation between perception and reasoning is clearly described and does not require complex training procedures.
2. The paper provides an interesting empirical observation that accurate localization can compensate for reduced image resolution in visual reasoning tasks.

### Weaknesses
1. The evaluation mainly compares SPARC with native inference and the “thinking with images” paradigm. However, several strong existing approaches such as DeepEyes, ViGoRL, or Pixel Reasoner are not directly evaluated. The paper also does not include a distilled SFT baseline (e.g., training directly on teacher reasoning traces). As a result, it is unclear how competitive the proposed pipeline is relative to stronger alternatives.
2. The experiments focus on relatively small models (4B and 8B). The framework relies on a manually structured perception–reasoning pipeline with explicit crop selection. The paper does not analyze whether the gains persist for larger or stronger VLMs, where perception capabilities may already be strong.
3. The method introduces a multi-stage inference pipeline with crop generation and additional prompts. While the paper reports reductions in visual token usage, it does not analyze end-to-end latency or runtime overhead. The practical impact of the pipeline is therefore unclear.

---

> ### Author Rebuttal · Authors · 2026-03-31
>
> We sincerely thank the reviewer for their constructive feedback and for highlighting the simplicity of our pipeline and the value of our empirical observation regarding localization versus resolution. We address the specific concerns below.
>
> ### Comparisons with Strong Baselines
>
> The comparison with strong baselines is indeed essential for a thorough evaluation. We'd like to direct the reviewer to Table 5 in the Appendix, where we have already provided a comprehensive comparison of SPARC against the aforementioned methods (DeepEyes, ViGoRL, and Pixel Reasoner) on the V* benchmark. In the revised manuscript, we will include a prominent cross-reference to this table from the main text.
>
> ### Distilled SFT Baseline
>
> The reviewer rightfully asked for a distilled SFT baseline (training directly on teacher reasoning traces). We initially excluded this based on the hypothesis that SFT distillation on our specific setup would lead to catastrophic forgetting. Nevertheless, as the reviewer's comment emphasizes, this should be validated empirically. We therefore conducted a pure distillation baseline using the same tuning recipe as SPARC SFT.
>
> As reported in the table below, the Distillation SFT overall performs worse than the zero-shot "Thinking with Images" (TwI) baseline, confirming our hypothesis. The table includes average performance on V*, HRBench-4K and HRBench-8K.
>
>
> |**Setting**|**256**|**512**|**Full**|
> |---|---|---|---|
> |**_Qwen 4B_**||||
> |TwI|36.8|52.2|73.2|
> |Distillation SFT|37.7|50.2|66.6|
> |SPARC SFT|**51.7**|**64.0**|**76.8**|
> |**_Qwen 8B_**||||
> |Twi|40.9|56.5|78.1|
> |Distillation SFT|42.7|54.9|72.1|
> |SPARC SFT|**53.1**|**64.3**|**82.4**|
>
>
> These results align with our hypothesis that Distillation SFT is not the right paradigm for this setting, where the data is synthetically generated and lacks broad diversity, particularly because of the complexity of the task requiring multi-turn tool calling. In addition, the base models have already undergone extensive SFT and RL alignment, and supervising the model to mimic teacher traces on a narrow dataset risks overwriting its general reasoning priors.
> Moreover, due to the interleaved long multimodal chains of thought, training the Distillation SFT baseline requires 4x the GPU memory at constant batch size compared to SPARC SFT.
> SPARC’s separation of perception and reasoning bypasses this degradation entirely while remaining efficient.
>
> ### Scaling to Larger Models
>
> To address the reviewer's request to evaluate scalability of our framework on larger models, we assessed the biggest available model of the Qwen family (235B-A22B) on the V* dataset (we were not able to evaluate the performance of Molmo 72B as the checkpoint still hasn't been released to the public).
>
>
> |**Setting**|**256**|**512**|**Full**|
> |---|---|---|---|
> |Native Baseline|47.5|52.1|89.5|
> |Thinking with Images|47.5|59.2|91.1|
> |SPARC|**53.0**|**65.1**|**91.6**|
>
>
> These results clearly demonstrate that SPARC generalizes beyond smaller-scale models. Even at the 235B scale, explicitly structuring the perception-reasoning pipeline and employing crop selection yields consistent performance improvements across all resolutions, outperforming also the "thinking with images" baseline.
>
> ### Efficiency Metrics
>
> We rely on token count as our primary efficiency metric because it is independent of hardware or model implementations. This makes it a reliable and reproducible metric for assessing computational overhead. In addition, it has the practical aspect of serving as the standard pricing metric for commercial inference APIs, which makes it the main concern for practitioners.
> Nevertheless, we understand and share the reviewer's interest in other metrics such as latency. To this end, we conducted a server-side Time to First Token (TTFT) analysis comparing SPARC and TwI. We note, however, that TTFT is a brittle, implementation-dependent, and hardware-bound metric that inherently fluctuates based on server load and framework-level optimizations.
>
> |**TTFT**|**SPARC**|**Twi**|
> |---|---|---|
> |**V***|||
> |256|0.2006|0.2846|
> |Full|2.9674|3.0494|
> |**_XLRS_**|||
> |256|0.0598|0.0744|
> |Full|13.1873|13.5052|
>
> As demonstrated in our analysis, SPARC achieves strictly lower latency than TwI across the board. Furthermore, given that SPARC at 256 resolution outperforms TwI in overall performance (IoU) at full resolution, the proposed pipeline provides higher accuracy while being over 200x faster in terms of TTFT.
>
> ### Multi-image and Video Reasoning
>
> While extending SPARC to multi-image or video reasoning is a compelling direction, it lies beyond the scope of this work. SPARC is explicitly designed to solve high-resolution visual reasoning bottlenecks within single images through 2D cropping. Extending this to video requires navigating the temporal dimension, which we view as a challenging and highly promising direction for future work. We will add a discussion regarding this to our conclusion section.

---

### Decision · Program_Chairs · 2026-04-30

**Decision:**

Accept (regular)

**Comment:**

The authors made a successful rebuttal, and most concerns have been solved. After reading the paper, reviewing comments, and the discussion, the AC tends to accept the paper.